# Extracting information on the spatial variability in erosion rate stored in detrital cooling age distributions in river sands

Jean Braun[1], Lorenzo Gemignani[2], and Peter van der Beek[3]

[1]GFZ German Research Centre for Geosciences, Telegrafenberg 14473, Potsdam, Germany
[2]Department of Earth Sciences, Vrije Universiteit Amsterdam, De Boelelaan 1083, 1081 HV Amsterdam, The Netherlands
[3] ISTerre, Université Grenoble Alpes, CS 40700, 38058 Grenoble Cedex 9, France

*Correspondence to:* J. Braun (jbraun@gfz-potsdam.de)

**Abstract.** One of the main purposes of detrital thermochronology is to provide constraints on regional scale exhumation rate and its spatial variability in actively eroding mountain ranges. Procedures that use cooling age distributions coupled with hypsometry and thermal models have been developed in order to extract quantitative estimates of erosion rate and its spatial distribution, assuming steady state between tectonic uplift and erosion. This hypothesis precludes the use of these procedures to assess the likely transient response of mountain belts to changes in tectonic or climatic forcing. Other methods are based on an a priori knowledge of the in-situ distribution of ages to interpret the detrital age distributions. In this paper, we describe a simple method that, using the observed detrital mineral age distributions collected along a river, allows to extract information about the relative distribution of erosion rates in an eroding catchment without relying on a steady-state assumption, the value of thermal parameters, or an a-priori knowledge of in-situ age distributions. The model is based on a relatively low number of parameters describing lithological variability among the various sub-catchments and their sizes, and only uses the raw ages. The method we propose is tested against synthetic age distributions to demonstrate its accuracy and the optimum conditions for it use. In order to illustrate the method, we invert age distributions collected along the main trunk of the Tsangpo-Siang-Brahmaputra river system in the Eastern Himalaya. From the inversion of the cooling age distributions we predict present day erosion rates of the catchments along the Tsangpo-Siang-Brahmaputra river system, as well as some of its tributaries. We show that detrital age distributions contain dual information about present-day erosion rate, i.e. from the predicted distribution of surface ages within each catchment and from the relative contribution of any given catchment to the river distribution. The method additionally allows comparing modern erosion rates to long-term exhumation rates. We provide a simple implementation of the method in Python code within a Jupyter Notebook that includes the data used in this paper for illustration purposes.

# 1  Introduction

Thermochronometric methods provide us with information pertaining to the cooling history of a rock. Various systems and minerals provide information on different parts of that cooling history, i.e. at a given temperature but more commonly within a range of temperatures. One of the main geological processes through which rocks experience cooling is exhumation towards the cold, quasi-isothermal surface (Brown, 1991). Young ages are commonly interpreted to indicate recent or rapid exhumation whereas old ages usually correspond to ancient or slow exhumation. Cooling ages can also record more discrete cooling events such as mineral crystallization during melt solidification, the nearby emplacement of hot intrusions (Gleadow and Brooks, 1979) or the rapid relaxation of isotherms at the end of an episode of rapid erosion (Kellett et al., 2013; Braun, 2016).

Besides collecting in-situ data, one can also collect and date a large number of mineral grains from a sand sample collected at a given location in a river draining an actively eroding area. Such detrital thermochronology datasets provide a proxy for the distribution of surface rock ages in a given catchment (Bernet et al., 2004; Brandon, 1992). By repeating this operation at different sites along a trunk stream, one obtains redundant information that can be used to document more precisely the spatial variability of in-situ thermochronological ages in a river catchment (Bernet et al., 2004; Brewer et al., 2006).

Methods have been devised to extract quantitative information from such detrital datasets concerning the erosion history of a tectonically active area, as well as estimates of its spatial variability. Ruhl and Hodges (2005) convolved their detrital age datasets with the hypsometry of the catchment to test the assumption of topographic steady-state in a rapidly eroding catchment of Nepalese Himalaya. Similarly, Stock et al. (2006) and Vermeesch (2007) combined detrital apatite (U-Th)/He age datasets with an age-elevation relationship established from in-situ samples to predict the distribution of present-day erosion rates in the eastern Sierra Nevada and White Mountains of California, respectively. Whipp et al. (2009) used simulations from a thermo-kinematic model to define the limits of applicability of such a technique, while Enkelmann and Ehlers (2015) used it in a glaciated landscape. Wobus et al. (2003, 2006) collected samples from tributaries of the Burhi Gandaki and Trisuli rivers to document the strong transition in erosion rate across a major topographic transition. By limiting their sampling to tributaries, they circumvented the need to develop and use a mixing model for the interpretation of their data. Brewer et al. (2006) derived optimal values for erosion rate in neighbouring catchments by comparing and mixing theoretical probability density distributions with detrital age data from the Marsyandi River in Nepal. Resentini and Malusà (2012) used a similar approach to interpret data from the Western Alps. McPhillips and Brandon (2010) used detrital cooling ages combined with in-situ age measurements to infer a recent increase in relief in the Sierra Nevada, California.

All of these methods rely on a-priori knowledge or hypotheses concerning the age distributions in the catchments drained by the river from which the samples have been collected. Here we explore the possibility of deriving first-order information about the spatial variability of erosion rate in the river catchment when no such knowledge exists and without making any assumption concerning the spatial distribution of ages. We propose to regard ages as passive markers (or tracers) that inform us on the proportion in which mixing takes place today, which must be directly proportional to the present-day erosion rate. For example, the most rapid present-day erosion rates should be predicted where the age distributions change the most rapidly along the river, everything else being accounted for, such as the relative size of neighboring sub-catchments or the potential

change in lithology between them. This knowledge about the distribution of erosion rates, which is obtained independently of the absolute values of the ages, can be used to estimate the distribution of ages in each sub-catchment drained by the river. This second piece of information provides additional insight into the spatial distribution of past cooling/erosional events.

In this paper, we present a method that relies on the raw age data only. This avoids any complication or bias that may arise from trying to compare the data to theoretical probability density distributions that rely on a thermal model prediction. We recognize the value of doing so, but thermal models require making assumptions about past geothermal gradient (heat flux), or rock thermal conductivity and heat production, which introduces additional uncertainty in interpreting the data. The first part of this paper describes the method. To demonstrate its accuracy and explore the limits of its applicability, we have applied the method to synthetic age datasets for which we know the erosion rate and its spatial variability. This is done in the second part of the paper. To illustrate the use of our method, we have applied it to a dataset collected in the Himalaya (along the Tsangpo-Siang-Brahmaputra river system). This is explained in the third part of the paper. There we show that the method yields reliable estimates of the distribution of present-day erosion rates in these areas as well as independent information on the spatial extent of past geological events. We conclude by suggesting potential ways in which the method could be improved. Note that the approach proposed here has been used on a set of detrital age distribution collected along the Inn River in the Eastern Alps (Gemignani et al., 2017). Gemignani et al. (2017) showed how age distributions characterized by young age peak likely produces high estimates of present day erosion rates when compared with catchments that contain older age peaks.

## 2 The method

### 2.1 Basic assumptions

We assume that we have collected a series of age datasets measured at $M$ specific points (or sites) along a river that drains a tectonically active regions where erosion rate is likely to vary spatially. We also assume that the datasets have been used to construct age distributions decomposed into $N$ age *bins* (see Figure 1) that may, for example, correspond to given, known geological events or, alternatively, have been selected after constructing a Kernel Density Estimate of the data (Vermeesch, 2012) and applying a mixture model to infer potential discrete age peaks in the age distribution (Sambridge and Compston, 1994). Although each bin corresponds to an age range, it might be easier to refer to it as representative of a given "age", which can be taken as the mean age of the range, for example. We will call $H_i^k$ for $k = 1, \cdots, N$ and $i = 1, \cdots, M$ the relative height of bin $k$ in distribution $i$. Because these are relative contributions, we have:

$$\sum_{k=1}^{N} H_i^k = 1, \quad \text{for all } i = 1, \cdots, M \tag{1}$$

The landscape is divided into *exclusive* contributing areas for each of the points along the main river where we have measured a dataset and compiled a distribution from it. We take the convention that Area 1 (of surface area $A_1$) is the area contributing to site 1, whereas Area 2 (of surface area $A_2$) is the area contributing to site 2 but not to site 1. Area $i$ (of surface area $A_i$) therefore contributes to site $i$ but not to the previous $i - 1$ sites (see Figure 2). In each Area $i$, we will assume that $\alpha_i$ is the

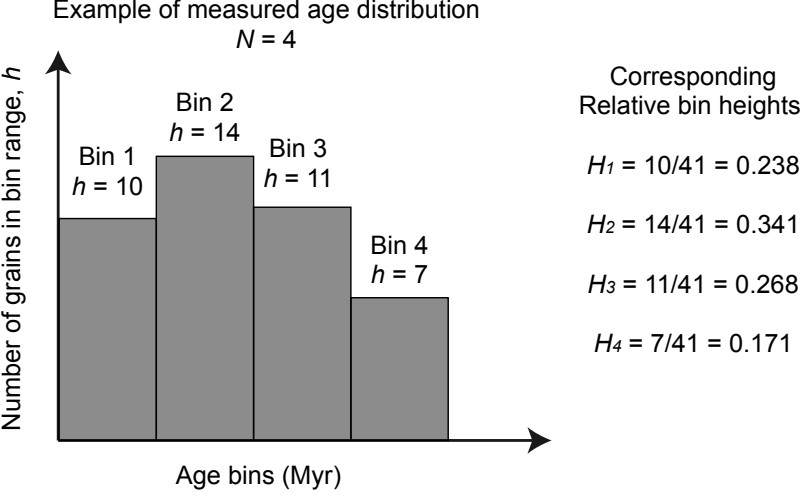

**Figure 1.** Example of a measured age distribution and the relative heights $H_i^k$ of the corresponding bins ($N = 4$ in this example).

relative abundance of the mineral used to estimate the age distribution in rocks being eroded from the surface. We will call $\alpha_i$ the "mineral concentration factor" of Area $i$. We take the convention that $0 < \alpha_i < 1$, with $\alpha_i = 1$ corresponding to an area $i$ with surface rocks that contain the mineral in abundance and $\alpha_i = 0$ corresponding to an area $i$ with surface rocks that do not contain the mineral. We also call $\epsilon_i$ the unknown present-day mean erosion rate in Area $i$.

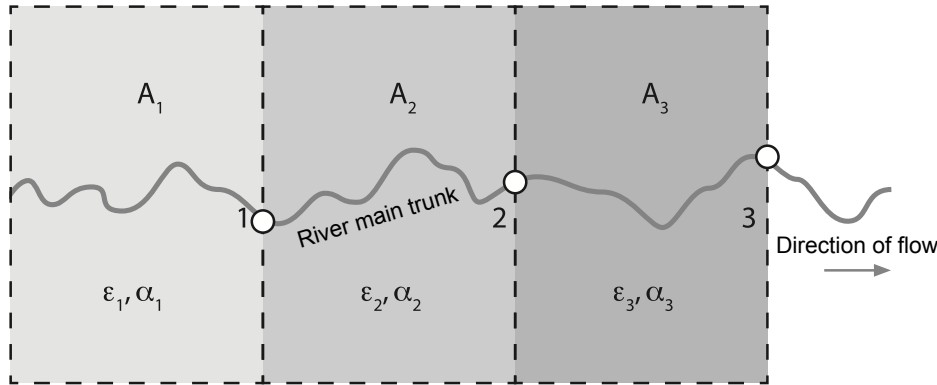

**Figure 2.** Schematic representation of how the landscape is divided into exclusive contributing areas $A_i$ (different shades of grey) for each of the points (here the circles labeled $i = 1, \cdots, 3$) along the main river where we have age distributions. $\epsilon_i$ and $\alpha_i$ are the assumed mean erosion rate and mineral concentration factor of Area $i$, respectively.

5      The surface areas, $A_i$, can be computed from a Digital Elevation Model. The value of the concentration factors depends on the regional geology. First-order estimates of the $\alpha_i$ parameters can be derived by considering the relative occurrence of the

specific mineral bearing rocks in each area from a geological map. However, recent work shows that mineral abundance in river sediments can vary significantly between tectonic units with similar lithology (Malusà et al., 2016). In their work, Malusà et al. (2016) propose a quantitative approach that can be used to infer potential "mineral fertility" bias between adjacent catchments and provide an example from the Alps. It is not the purpose of this paper, however, to speculate on the bias induced by the mineral fertility parameter in the interpretation of detrital age distributions. We refer to previous work such as that of Malusà et al. (2016) or Resentini and Malusà (2012) on the subject and assume that if the necessary data is available to perform a correction for this bias, it should be made by adjusting the value of the mineral concentration factor $\alpha_i$ accordingly.

From these simple assumptions, we can write that the number of grains of age $k$ coming out of catchment $i$ is given by:

$$D_i^k = A_i \alpha_i \epsilon_i C_i^k = F_i \epsilon_i C_i^k \qquad (2)$$

where $F_i = A_i \alpha_i$ and $C_i^k$ is the unknown relative concentration of grains of age $k$ in surficial rocks in Area $i$. We also have:

$$\sum_{k=1}^{N} C_i^k = 1, \quad \text{for all } i \qquad (3)$$

because the $C_i^k$ are also relative or normalized concentrations. The relative concentrations, $C_i^k$ tells us to what extent age $k$ has been preserved in surficial rocks of Area $i$, whereas $\epsilon_i$ is a measure of present-day erosion rate in Area $i$.

## 2.2 Downstream bin summation along main trunk

We can now write that the predicted height of bin $k$ in the distribution observed at site $i$ should be equal to the total number of grains of age-bin $k$ coming from all upstream areas divided by the total number of grains of all ages coming from all upstream areas:

$$H_i^k = \left(\sum_{j=1}^{i} D_j^k\right) / \left(\sum_{k=1}^{N}\sum_{j=1}^{i} D_j^k\right) = \left(\sum_{j=1}^{i} F_j \epsilon_j C_j^k\right) / \left(\sum_{k=1}^{N}\sum_{j=1}^{i} F_j \epsilon_j C_j^k\right) \qquad (4)$$

We can slightly re-arrange this to obtain:

$$H_i^k = \left(\sum_{j=1}^{i} F_j \epsilon_j C_j^k\right) / \left(\sum_{j=1}^{i} F_j \epsilon_j \sum_{k=1}^{N} C_j^k\right) = \left(\sum_{j=1}^{i} F_j \epsilon_j C_j^k\right) / \left(\sum_{j=1}^{i} F_j \epsilon_j\right) \qquad (5)$$

If we divide the numerator and denominator of this expression by $F_1 \epsilon_1$, we obtain:

$$H_i^k = \sum_{j=1}^{i} \rho_j C_j^k / \sum_{j=1}^{i} \rho_j \qquad (6)$$

where:

$$\rho_j = \frac{F_j \epsilon_j}{F_1 \epsilon_1} \qquad (7)$$

is the contribution from Area $j$ relative to Area 1. Note that, if we assume that we can confidently estimate $F_j = A_j \alpha_j$, $\rho_j$ becomes a measure of the unknown erosion rate, $\epsilon_j$, in Area $j$ relative to the unknown erosion rate, $\epsilon_1$, in Area 1.

## 2.3 Incremental formulation

We now express Equation (6) as an incremental relationship between $H_i^k$ and $H_{i-1}^k$ only, i.e. between the relative bin heights of distributions measured at two successive points along the main trunk. From Equation (6), we can write:

$$H_i^k = \sum_{j=1}^{i} \rho_j C_j^k / \sum_{j=1}^{i} \rho_j = \left(\sum_{j=1}^{i-1} \rho_j C_j^k + \rho_i C_i^k\right) / \left(\sum_{j=1}^{i-1} \rho_j + \rho_i\right) \tag{8}$$

and by dividing numerator and denominator by $\sum_{j=1}^{i-1} \rho_j$, we obtain:

$$H_i^k = \left(H_{i-1}^k + \delta_i C_i^k\right) / \left(1 + \delta_i\right) \tag{9}$$

where:

$$\delta_i = \rho_i / \sum_{j=1}^{i-1} \rho_j \tag{10}$$

is the contribution of Area $i$ relative to the contribution of all upstream Areas, i.e., what sediment is entering the river from Area $i$ relative to what is already in the river. We will call $\delta_i$ the "relative contribution" from Area $i$

We can also write:

$$H_i^k - H_{i-1}^k = (C_i^k - H_i^k)\delta_i, \quad \text{for } i = 1, \cdots, M \text{ and } k = 1, \cdots, N \tag{11}$$

From this relationship we see that the relative changes in bin height between two successive sites along the main stream contain information about the present-day erosion rate in the intervening catchment, through the relative contributions $\delta_i$ and the relative concentrations in surface bedrock in Area $i$, $C_i^k$. If one knows either $\delta_i$ or the $C_i^k$ (for $k = 1, \cdots, N$), one can derive the value of the other quantity(ies). For example Resentini and Malusà (2012) assumed that they knew the values of the $C_i^k$ surface concentrations in each sub-catchment to derive estimates of the corresponding $\delta_i$. Here we will try to assess whether the values of both the relative surface concentrations $C_i^k$ and the relative contributions $\delta_i^k$ can be estimated.

## 2.4 Estimating erosion rates

Considering that we have $M$ sites along the main river trunk and that we have selected to use $N$ bins to describe the distributions of ages, we have $N \times M$ unknown values for the relative heights of the bins describing the distribution of ages in the source areas (the $C_i^k$) and another set of $M - 1$ unknown values for the relative contributions $\delta_i$. However, we only have $N \times M$ equations (see eq. 11) and the problem is clearly underdetermined, i.e. we have more unknowns than equations among the unknowns. This means that there is an infinite number of solutions that satisfy the equations. We cannot estimate the values of all unknowns, but, as we will show now, we can estimate bounds on the value of the unknowns $\delta_i$.

Two cases must be considered. First, if there is a noticeable change in relative bin heights in the detrital record between sites $i-1$ and $i$, i.e. $H_i^k \neq H_{i-1}^k$, the distribution of ages in Area $i$, $C_i^k$, must be different from that in the river at site $i$, i.e. $C_i^k \neq H_i^k$ and, consequently, the relative contribution from Area $i$, $\delta_i$ must be finite, i.e. $\delta_i > 0$ and therefore $\epsilon_i > 0$. This means that

there must be a minimum finite value for $\delta_i$ and therefore for $\epsilon_i$ to explain the difference in distribution between sites $i-1$ and $i$. We will call this minimum but finite value $\delta_i^m$ (and $\epsilon_i^m$, respectively). There is another solution to consider where $\delta_i \to \infty$ and $C_i^k \to H_i^k$, which is correct regardless of the values of the $H_{i-1}^k$ and which corresponds to the situation where the relative contribution from Area $i$ is so large that it completely overprints the river signal. We thus have two bounds for the relative contributions $\delta_i$ at each site, one finite and the other infinite:

$$\delta_i \in [\delta_i^m, \infty] \tag{12}$$

The second case to consider is when the relative bin heights between two successive sites do not change , i.e. $H_i^k = H_{i-1}^k$ for all bins $k$. In this case, we cannot tell if this is because the erosion rate in catchment $i$ is nil ($\epsilon_i = 0 \to \rho_i = 0 \to \delta_i = 0$), or because the signature of the source in catchment $i$, i.e. the distribution of ages at the surface, is identical to that of the river ($C_i^k = H_i^k = H_{i-1}^k$), and we have no constraints on $\delta_i$ or the erosion rate in Area $i$. Although this situation may arise, we will now only consider the case where $H_i^k \neq H_{i-1}^k$ and try to find the value of $\delta_i^m$.

For this, we re-write equation 11 as:

$$C_i^k = \frac{H_i^k - H_{i-1}^k}{\delta_i} + H_i^k \tag{13}$$

Considering that the $H_i^k$ are relative bin heights, i.e. $H_i^k \in [0,1]$ and $\sum_{k=1}^N H_i^k = 1$, implies that the $C_i^k$ are also relative bin heights. Therefore we must have $\sum_{k=1}^N C_i^k = 1$ and $C_i^k \in [0,1]$. This leads to two constraints on the value of the relative contribution $\delta_i$:

$$C_i^k > 0 \to \delta_i > \max_{k=1,\cdots,N} \frac{H_{i-1}^k - H_i^k}{H_i^k} \tag{14}$$

and:

$$C_i^k < 1 \to \delta_i > \max_{k=1,\cdots,N} \frac{H_i^k - H_{i-1}^k}{1 - H_i^k} \tag{15}$$

for all $i = 1, \cdots, M$. The first condition applies where there is a decrease in any relative bin height $k$ between site $i-1$ and site $i$, i.e. $H_i^k < H_{i-1}^k$, whereas the second condition applies where there is an increase in any relative bin height $k$ between locations $i-1$ and $i$, i.e. $H_i^k > H_{i-1}^k$. We can therefore conclude that the minimum values of the contribution factors that are necessary to explain the change in relative bin size from site to site are given by:

$$\delta_i^m = \max_{k=1,\cdots,N} \left( \frac{H_{i-1}^k - H_i^k}{H_i^k}, \frac{H_i^k - H_{i-1}^k}{1 - H_i^k} \right) \quad \text{for } i = 1, \cdots, M \tag{16}$$

From these estimates of the contribution factors, $\delta_i^m$, we can then derive the value of the corresponding erosion rate in each Area $i$, $\epsilon_i^m$, that is necessary to explain the observed variations in age distributions along the main river trunk, by using the following relationship that we derive in the appendix:

$$\epsilon_i^m = \frac{F_1 \epsilon_1^m}{F_i} \delta_i^m \prod_{j=1}^{i-1} (1 + \delta_j^m) \quad \text{for } i = 1, \cdots, M \tag{17}$$

assuming that $\delta_1^m = 0$ and $\epsilon_1^m = 1$ such that the estimated minimum erosion rates are relative erosion rates scaled by the unknown erosion rate in the first catchment.

From the values of the minimum contribution factors, $\delta_i^m$, we can also estimate the relative surface concentrations of each age bin $k$ in each catchment $i$, using:

$$5 \quad C_i^k = \frac{H_i^k - H_{i-1}^k}{\delta_i^m} + H_i^k \tag{18}$$

## 3 Using age distributions from tributaries

Age distributions from tributaries can be included to improve the solution locally, i.e. in the tributary catchment. Let's call $A_T$, $\alpha_t$ and $\epsilon_T$ the catchment area, the mineral concentration factor and the mean erosion rate of the tributary catchment, and $A_M$, $\alpha_M$ and $\epsilon_M$ the catchment area, the mineral concentration factor and the mean erosion rate of the rest of catchment $i$.

10      For each bin $k$ in the catchment $i$, we can write:

$$F_i \epsilon_i^m C_i^k = F_T \epsilon_T C_T^k + F_M \epsilon_M C_M^k \tag{19}$$

where $F_T = A_T \alpha_T$ and $F_M = A_M \alpha_M$. By conservation of eroded rock mass, we have:

$$F_M \epsilon_M = F_i \epsilon_i^m - F_T \epsilon_T \tag{20}$$

which we can use to transform Equation (19) into:

$$15 \quad F_i \epsilon_i^m C_i^k = F_T \epsilon_T C_T^k + (F_i \epsilon_i^m - F_T \epsilon_T) C_M^k \tag{21}$$

to obtain:

$$C_M^k = \frac{F_i \epsilon_i^m C_i^k - F_T \epsilon_T C_T^k}{F_i \epsilon_i^m - F_T \epsilon_T} \tag{22}$$

Using the method for the main trunk data described in the previous sections, we know $\epsilon_i$ and $C_i^k$. The tributary data (age distributions) gives us the $C_T^k$ surface concentrations as measured in the tributary stream (i.e. $C_T^k = H_T^k$) and we can solve for

20     the surface concentrations $C_M^k$ assuming first that the erosion rate is uniform in the catchment $i$, i.e. $\epsilon_T = \epsilon_M = \epsilon_i$, to give:

$$C_M^k = \frac{F_i C_i^k - F_T C_T^k}{F_i - F_T} \tag{23}$$

However, this may lead to unrealistic values of the relative surface concentrations $C_M^k$, i.e. not comprised between 0 and 1. Consequently, two conditions need to be added so that $0 < C_M^k < 1$ for all $k$. The first condition ($C_M^k > 0$) yields:

$$\epsilon_T < \frac{F_i C_i^k}{F_T C_T^k} \epsilon_i^m \tag{24}$$

25     while the second condition ($C_M^k < 1$) yields:

$$\epsilon_T < \frac{F_i (1 - C_i^k)}{F_T (1 - C_T^k)} \epsilon_i^m \tag{25}$$

The true erosion rate must satisfy both conditions and we therefore select the smallest value of $\epsilon_T$ obtained by considering any relative surface concentration difference between the tributary sub-catchment concentration ($C_T^k$) and that of the entire catchment ($C_i^k$).

## 3.1 Uncertainty estimates by bootstrapping

We assess the robustness of our estimates of minimum erosion rate $\epsilon_i^m$ and corresponding relative concentrations $C_i^k$, derived from finite size samples by bootstrapping. For this, we simply use the method described above on a large number of sub-samples of the observed distributions constructed by random sampling of the observed distributions with replacement. This yields distributions of erosion rate and relative concentrations that can be used to estimate the uncertainty arising from the finite sample size. These distributions are usually not normal and we use their median value, rather than their mean, as the most

likely estimate of erosion rate and their standard deviation to represent uncertainty.

The code is provided as a Jupyter Notebook containing python code and explanatory notes that refer to the equations given in this manuscript. The user must provide a series of input files containing (a) the description of the sites, i.e. the order in which the sites are located along the river, whether they drain into the main river stem or into a tributary, the drainage area $A$, the mineral concentration factor $\alpha$, (b) the bin sizes and (c) the observed age data at each site. The code produces distributions of

estimates of erosion rate and relative concentration of grains of ages within each bin for each site from the bootstrapping.

## 4  Assessing the method on synthetic distributions

We have assessed the reliability of the erosion rate estimates obtained from the method by applying it to synthetic age distributions made of $N = 4$ age bins at $M = 5$ sites. We have assumed known mineral concentration factors, $\alpha_i$, contributing areas, $A_i$, and relative erosion rates. To construct the distributions, we created very large samples ($10^6$ grains) from normal

distributions of grains having ages centered on 4 peak values (20, 40, 60 and 80 Myr) with a standard deviation of $\Delta a = 5$ Myr. The amplitude of the peaks is first set to $(0, 0, 0, 1)$, $(0, 0, 1, 0)$, $(0, 1, 0, 0)$, $(1, 0, 0, 0)$ and $(0, 0, 0.5, 0.5)$, for the five sites. In this way, we are likely to maximize relative bin height differences between two consecutive sites. We will later relax this assumption and see how it impacts the estimates obtained by the method. We then constructed 5 grain distributions corresponding to each of the sites by adding the original distributions in proportions given by the relative contributions (product of area, mineral

concentration factor and erosion rate) of each area. We then sampled these distributions by randomly selecting $n = 100$ grains. This number was chosen because, in many detrital age studies, it represents the average number of grains collected at any given site. We applied our method to these synthetic datasets to obtain distributions of minimum erosion rate by bootstrapping and took the median value as a reliable estimate of the minimum erosion rate in each area necessary to satisfy the synthetic age distributions. We repeated this operation 1000 times (i.e. by generating 1000 synthetic datasets) to obtain the distributions

of estimates of minimum erosion rates and compared them to the imposed erosion rates used to construct the synthetic age distributions. In a first set of four experiments, the areas and mineral concentration factors are identical for each site. In the first experiment, the erosion rates are chosen to vary greatly (by up to two orders of magnitude) between consecutive sites. In

the second experiment, all sites have the same erosion rate, in the third experiment, the erosion rate increases by a factor two from site to site and in the fourth experiment, the erosion rate decreases by a factor two from site to site. The results are shown in Figure (3).

We see that the estimated erosion rates (median value) are in good agreement with the imposed (true) erosion rates, especially when large jumps in erosion rate exists between successive sites/catchments. However, in some cases, there appears to be an artificial increase in estimated erosion rate from site to site. This is clearly seen in the case where the imposed erosion rate is assumed uniform at all sites or to decrease from site to site. In both cases, the method predicts an apparent (or spurious) increase in erosion rate at the last site.

To determine what controls the reliability of these estimates, we performed two other sets of experiments. The first assumes random amplitudes for the peaks in each of the catchment (Figure 4) and the seconds assumes that the $F_i$ (product of drainage area $A_i$ by relative abundance $\alpha_i$) increase downstream (Figure 5). When the amplitudes of the peaks are random, the differences in peak height between two successive sites is much smaller, leading to deteriorated values for the estimated erosion rates. Oppositely, when the contributing areas (or relative abundances) increase downstream, the estimates are improved.

Consequently, the accuracy of the estimates of erosion rate obtained from our method rely on whether the two successive age distributions used to estimate the $\delta_i^m$ are markedly different and whether the size of the two successive catchments increases or, at least, does not decrease substantially. These conditions should be kept in mind when interpreting the results obtained by our method.

## 5 Applications to a detrital age dataset

To illustrate the method, we now apply it to a detrital age dataset from the Eastern Himalaya that contains ages obtained using the muscovite $^{40}$Ar/$^{39}$Ar thermochronometer. The data were combined from published age datasets collected along the main trunk of the Tsangpo-Siang-Brahmaputra river system, as well as along some of its tributaries (Figure 6), using age bins given in Table 1. Samples A,B,C (composite sample), X and Y are from Lang et al. (2016) and samples Z, T-40a and T-41a are from Bracciali et al. (2016). The complete age datasets are given in the Data Repository, Table S1. In Table 2, we give the relative position of the successive samples along the main trunk of the river, $i$, the respective exclusive contributing areas, $A_i$ and the mineral concentration factors, $\alpha_i$. We first used constant values of 1% (fourth column in Table 2). We will also use variable values for the mineral concentration factors, making the simple assumption that surficial rocks have an average mineral composition that depends mostly on their lithology. A very simple way to proceed is to look at the surface lithology indicated in the geological map of the region for each studied catchment. In this case we used the geologic map of the Eastern Himalaya from Yin et al. (2010). The semi-quantitative $\alpha$-values computed for each catchments are given in the fifth column of Table 2. We are aware that, for such a complex and vast area, this approximation can yield to poor estimates of the concentration of the target mineral in surface rocks. For more accurate methods (that also require additional data), we refer the user to published work by others that have investigated this issue by looking, for instance, at petrographic and heavy mineral density in modern

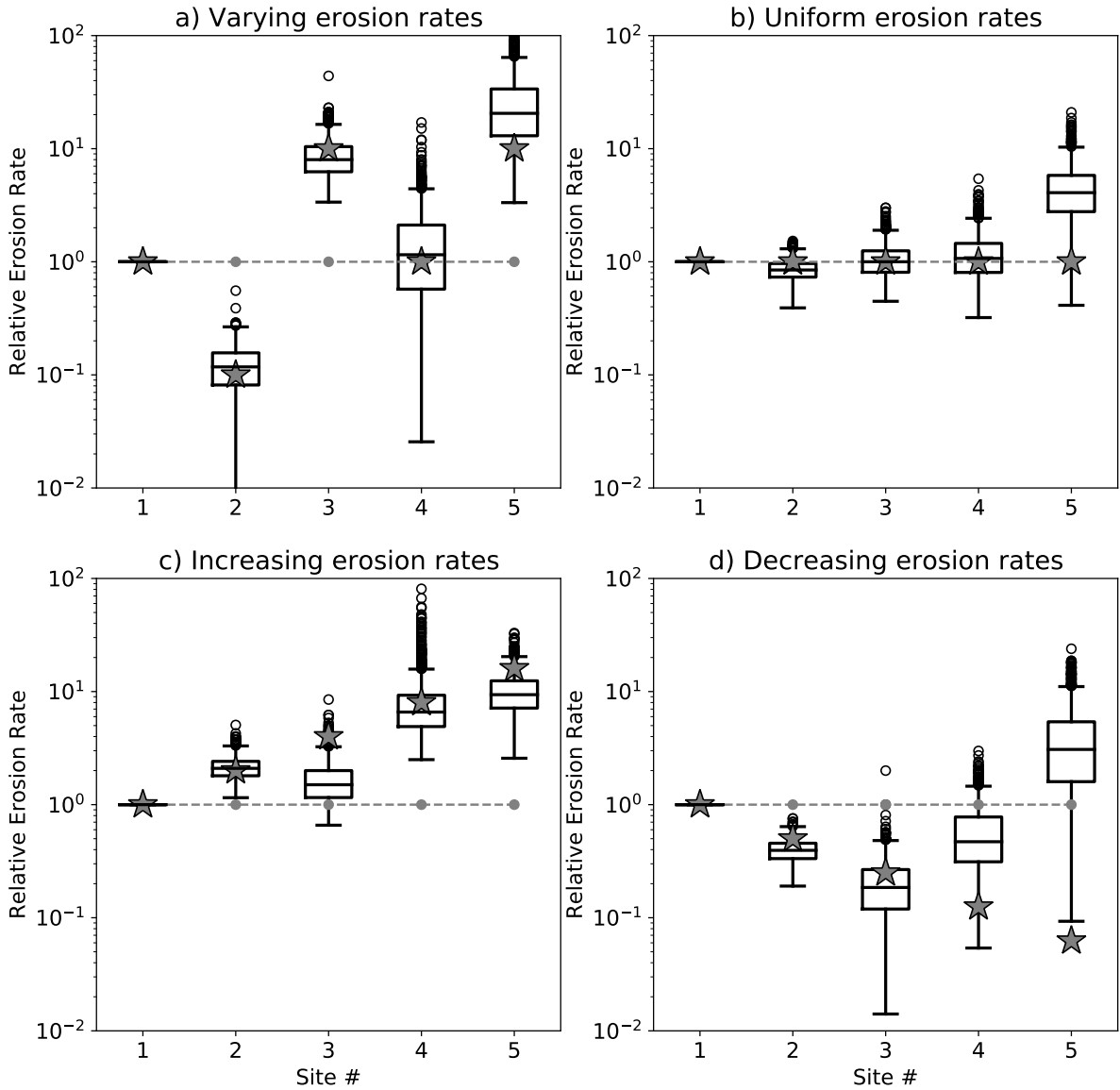

**Figure 3.** Results of the method applied to the first set of synthetic datasets. Computed erosion rate distributions for four synthetic datasets. For each distribution, the box extends from the lower to upper quartile values, the line corresponds to the median value and whiskers extend from the box to show the range of the erosion rate estimates, excluding outliers. Outliers are indicated by small circles past the end of the whiskers. For each site, the grey stars correspond to the imposed erosion rates and the dashed grey line gives the product of the imposed area and fertility factors, $F_i = A_i \alpha_i$.

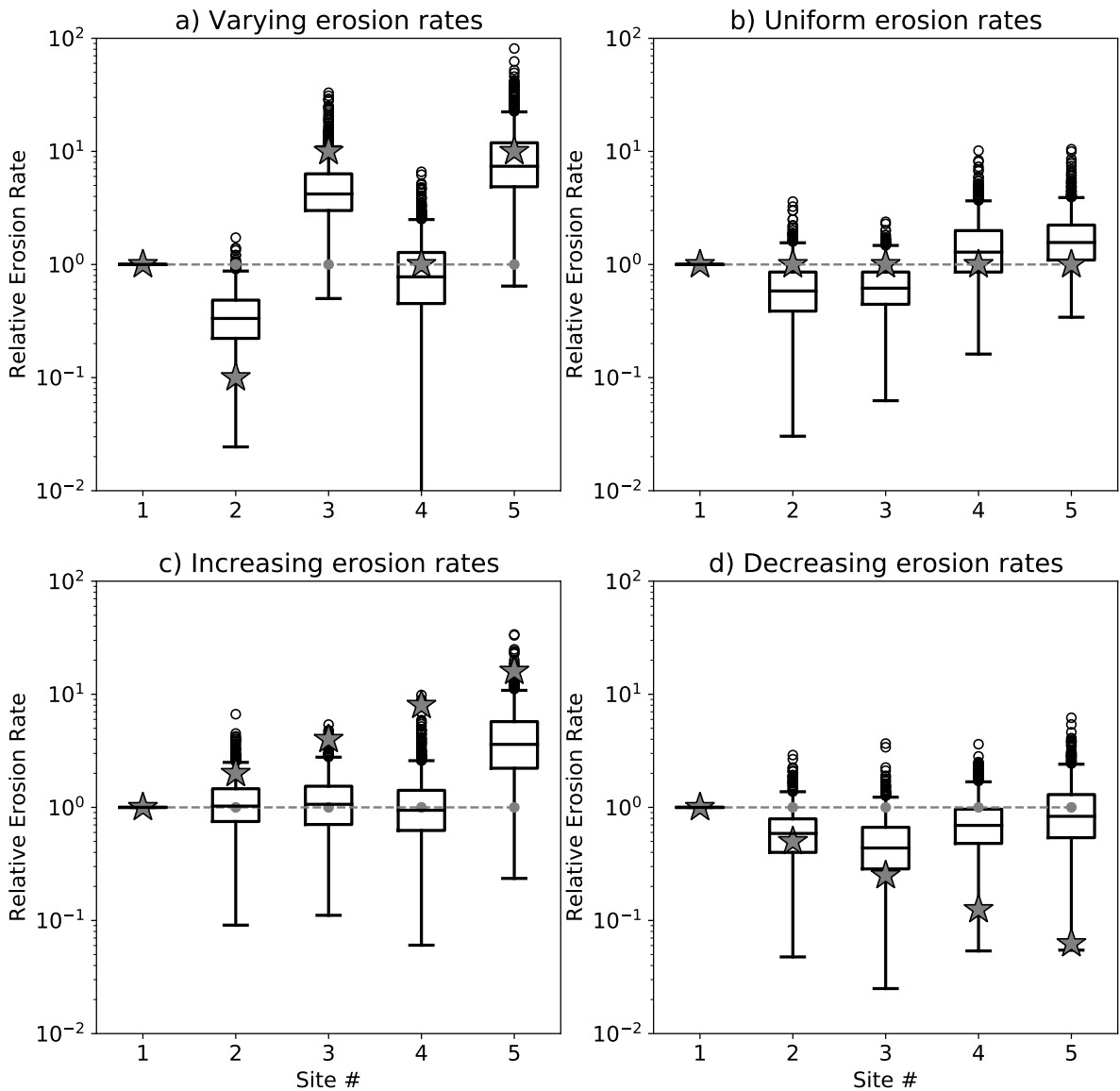

**Figure 4.** Results of the method applied to the second set of synthetic datasets with random peak amplitudes. See Figure 3 caption for further details.

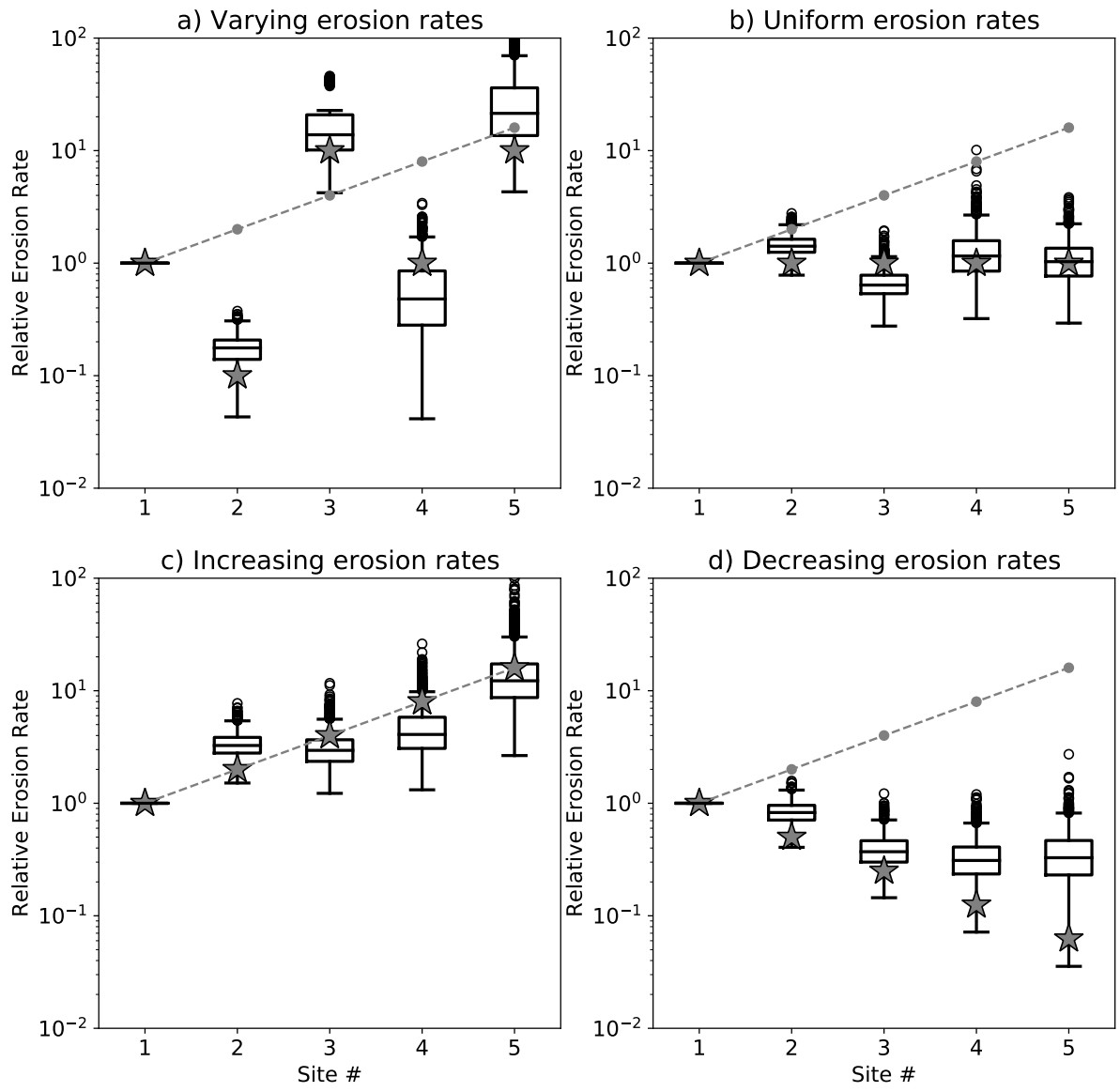

**Figure 5.** Results of the method applied to the second set of synthetic datasets with increasing $F_i = A_i \alpha_i$ downstream. See Figure 3 caption for further details.

river sediments such as SRD index (Garzanti and Andò, 2007) or the method proposed to compute mineral "fertility" (Malusà et al., 2016).

We will investigate the effect of using variable concentrations of muscovite-bearing rocks in each of the catchments as derived from a geological map of the area (fifth column in Table 2).

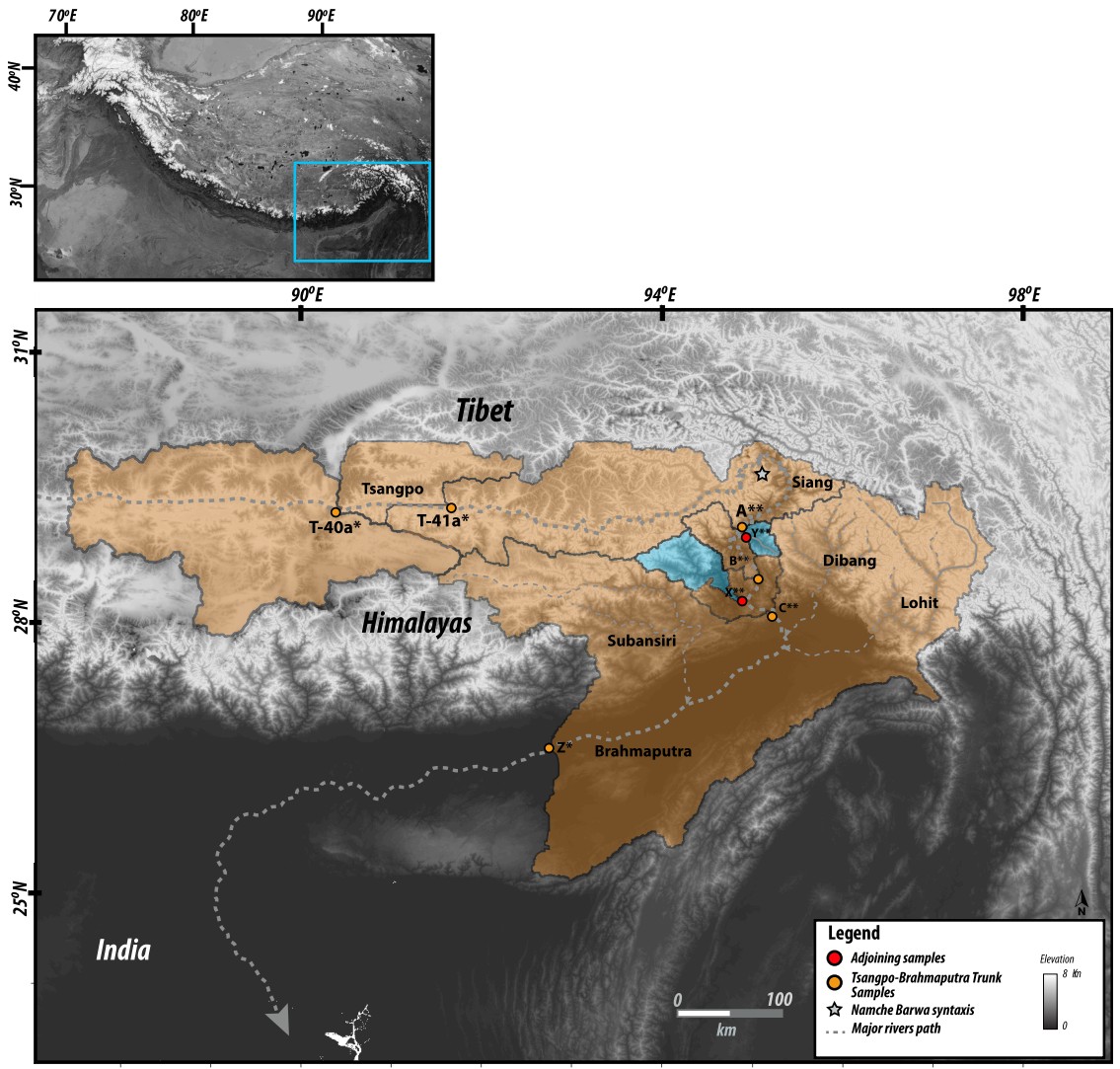

**Figure 6.** a) Location of the study area and b) location and name of sampling sites and geometry of the drainage basins contributing to each site. The orange shading represents catchments draining directly into the main trunk; pale blue shading represents the tributary catchments or sub-catchments.

**Table 1.** Age bins used to construct age distributions shown in Figure 8 and used in our example.

| Bin 1 | Bin2 | Bin 3 | Bin 4 | Bin 5 |
|---|---|---|---|---|
| 0-5 Ma | 5-10 Ma | 10-20 Ma | 20-50 Ma | 50-500 Ma |

**Table 2.** Relative position along the main trunk of the Tsangpo-Siang-Brahmaputra river system. Negative numbers indicate samples collected along a tributary. Catchment areas and mineral concentration factors used to compute the erosion rate reported in Table 7. The two columns correspond to two different sets of values used for comparison. Site names refer to locations shown in Figure 6.

| Site | Position | Catchment area (km$^2$) | mineral concentration factor 1 | mineral concentration factor 2 | Reference |
|---|---|---|---|---|---|
| TG-40a | 1 | 55395 | 0.01 | 0.03 | Bracciali et al. (2016) |
| TG-41a | 2 | 13265 | 0.01 | 0.03 | Bracciali et al. (2016) |
| A | 3 | 41374 | 0.01 | 0.02 | Lang et al. (2016) |
| Y | -4 | 1250 | 0.01 | 0.25 | Lang et al. (2016) |
| B | 5 | 2092 | 0.01 | 0.09 | Lang et al. (2016) |
| X | -6 | 2135 | 0.01 | 0.18 | Lang et al. (2016) |
| C | 7 | 1451 | 0.01 | 0.12 | Lang et al. (2016) |
| Z | 8 | 111706 | 0.01 | 0.30 | Bracciali et al. (2016) |

Results are shown in Figure (7) as distributions of computed relative minimum erosion rates (i.e. normalized such that the mean erosion rate is 1) obtained by bootstrapping. The computed surface concentrations, $C_i^k$, for each site $i$ are shown in Figure (8). Figure (9) contains maps of the various catchments shaded according to their predicted median erosion rate and surface concentrations of grains of age within each range. Predicted concentrations are scaled such that the sum of the five age bin concentrations is 1 in each catchment. We see that predicted minimum erosion rates increase by about two orders of magnitude with distance along the main river trunk from its source area along the southern margin of the Tibetan Plateau. Maximum erosion rates are observed in catchment C (and sub-catchment X) that is closest to the eastern Himalayan syntaxis. The amplitude of the jumps in erosion rate between sites A and B, and B and C are potentially amplified by our method because sites A, B and C have relatively small areas, $A_i$ and contains relatively uniform distributions of ages among the five bins. As we have noticed in our synthetic examples, this may lead to a spurious increase in erosion rate. Further downstream (catchment Z), the predicted minimum erosion rate remains high but lower than observed near the syntaxis. This estimate is likely to be robust as site Z has a very large contributing area and has an age distribution that is markedly different from that of the previous catchment (C).

The most salient result predicted by the method is that the erosion rate in the easter Himalayan syntaxis should be at least 5-7 times higher than the mean erosion rate along the Tsangpo-Siang-Brahmaputra basin. This estimate is in good agreement with the conclusions of Stewart et al. (2008) who used U-Pb ages of detrital zircon grains from the Brahmaputra River to

**Table 3.** Predicted erosion rate in the successive sub-catchments obtained by assuming uniform values for the $\alpha_i$ and variable values derived from the relative abundance of muscovite-bearing surface rock in the geological map.

| Site | Erosion rate estimate using uniform $\alpha_i$ | Erosion rate estimate using variable $\alpha_i$ |
|------|:---:|:---:|
| TG-40a | 1 | 1 |
| TG-41a | 1.06 | 1.12 |
| A | 0.78 | 1.17 |
| Y | 5.29 | 1.54 |
| B | 5.39 | 1.98 |
| X | 36.4 | 8.71 |
| C | 36.4 | 8.77 |
| Z | 9.40 | 0.90 |

demonstrate that approximately half of the sediment flux carried by the Brahmaputra River originates from an area around the eastern syntaxis that represents only 2% of the total area of the river drainage basin. This implies that erosion rate in the vicinity of the syntaxis should be approximately 25 times higher than the mean erosion rate. Similar estimates were obtained by Enkelmann et al. (2011) using a larger detrital age dataset from the area, while a relatively smaller estimate (erosion rate

should only be 5 times higher than the mean in the syntaxis area) was obtained by Singh and France-Lanord (2002) using the isotopic composition of sediments collected along the Brahmaputra River. In Table 3, we compare the erosion rate estimates obtained using uniform surface concentration factors ($\alpha_i = 0.01$) with those obtained using variables values (given in the fifth column of Table 2). We note that although the values obtained for the sub-catchments upstream of the syntaxis (sites Y and B) are somewhat reduced, the very large relative erosion rates (5-8 times the entire basin average) predicted in the syntaxis (sites

X and C) is a robust outcome of the method.

We also note that erosion rates in sub-catchments Y and X do not need to be noticeably smaller than the estimates of erosion rate in their host catchments (B and C). As explained earlier, the true erosion rates could be larger than those of their host catchments. This could be the case for sub-catchment Y where we predict an erosion rate identical to that of catchment B. These estimates are likely to be reliable because their area is similar to that of their sub-catchment (they occupy a non-negligible

portion of their host catchment) and because they have strikingly different age distributions than their host catchments (B and C) (Figure 8). One can also see from the age distributions shown in Figure (8) how the large contribution from bin 4 at site Y affects the relative height of bin 4 at site B and, similarly, how the large contribution from bin 3 affects the relative height of bin 3 at site C.

Interestingly, there is a good correspondence between present-day erosion rate and where the youngest ages are being

generated (sites B and C), with the notable exception of the most downstream catchment (Z). In other words, where the mixing analysis predicts high minimum erosion rate to account for a substantial change in the age distribution between two adjacent

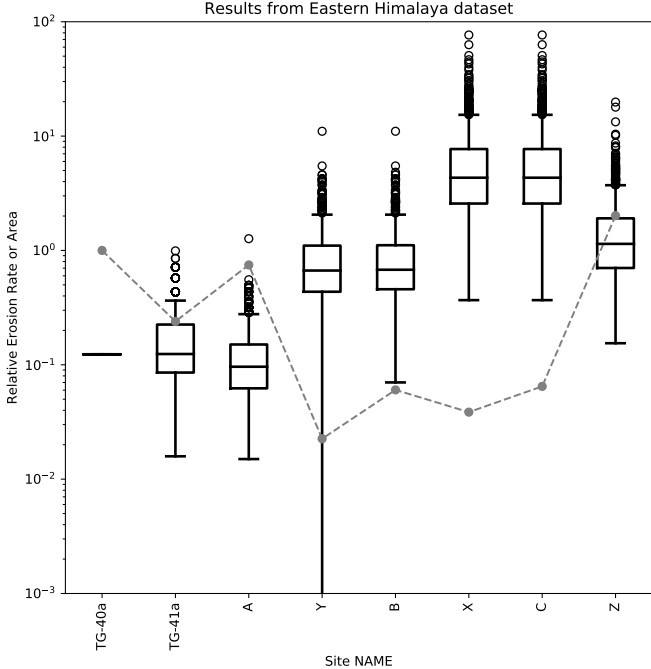

**Figure 7.** Computed erosion rate distributions obtained by applying the method to a Himalayan dataset. For each distribution, the box extends from the lower to upper quartile values, the line corresponds to the median value and whiskers extend from the box to show the range of the erosion rate estimates, excluding outliers. Outliers are indicated by small circles past the end of the whiskers. Erosion rate values are normalized such that the mean is 1. The grey circles connected by a dashed grey line are the product of the imposed area and mineral concentration factors, $F_i = A_i \alpha_i$ at each site. Site names refer to locations shown in Figure 6.

catchments, is also where it predicts the highest concentration of young ages in the surface rocks. At the downstream end of the river (Catchment Z), we predict a relatively high minimum erosion rate from the mixing model but a relatively low concentration of young ages in comparison to the other catchments. This could mean that, in catchment Z, the present-day high erosion rate is relatively recent and has not led yet to a complete resetting of cooling ages which were set during earlier events.

5    We also note that the difference between the two quartile values (vertical size of the boxes in Figure 7 and 8) is large, of the order of 50-100% of the median value of predicted erosion rate values. This indicates that our method can only provide order-of-magnitude estimates of the minimum erosion rate necessary to explain the age distributions. Interestingly, the difference between the two quartile values does not increase downstream, which demonstrates that the uncertainty introduced by using incomplete or non-representative sub-samples of the true distributions at each of the station does not accumulate as our

10   algorithm proceeds from station to station. This results from the incremental nature of our algorithm, as shown by Equation 11.

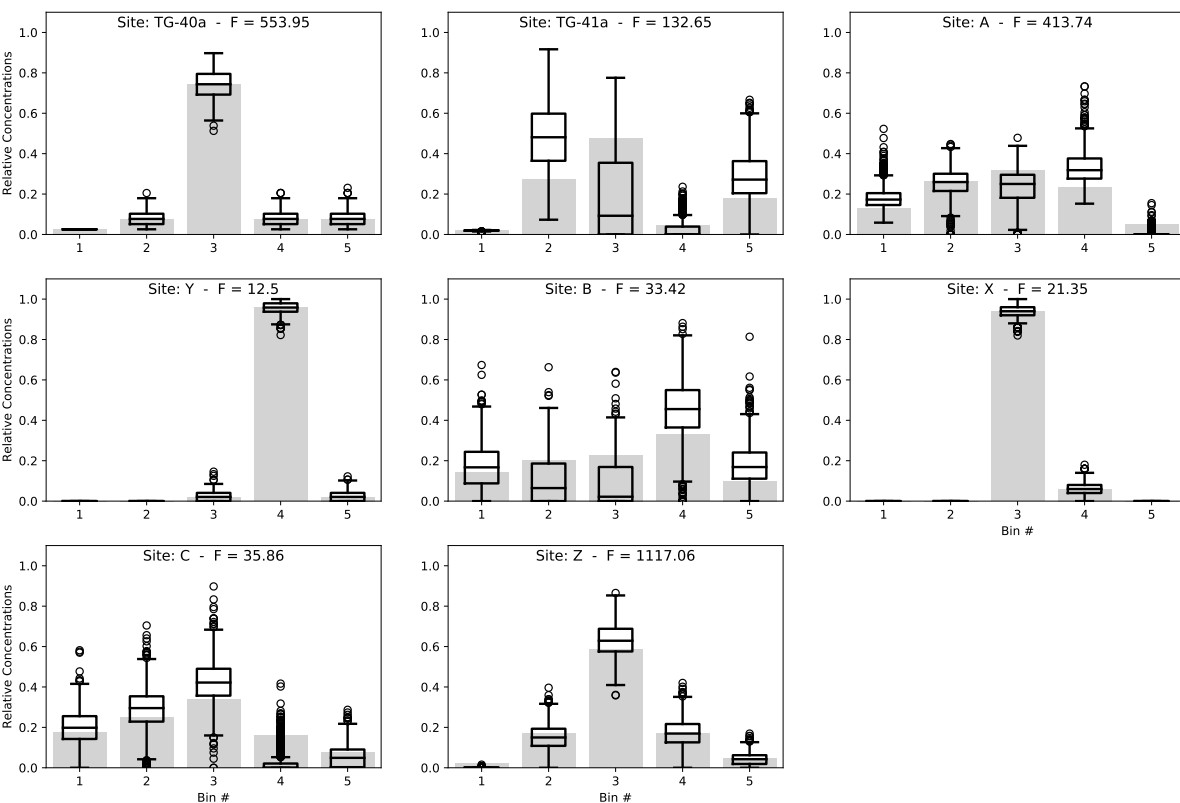

**Figure 8.** Observed distributions of ages (light grey bars) in samples collected at sites shown in Figure 6 on which the distributions of predicted surface age distributions have been superimposed for each site. For each distribution, the box extends from the lower to upper quartile values, the line corresponds to the median value and whiskers extend from the box to show the full range, excluding outliers. Outliers are indicated by small circles past the end of the whiskers.

## 6 Mineral concentration factors and their uncertainty

One of the sources of uncertainty in our estimates of the erosion rate comes from the assumed value of the mineral concentration factors, $\alpha_i$, which might be difficult to estimate in many situations (Malusà et al., 2016). We can compute the uncertainty on the erosion rates, $\Delta\epsilon_i$ arising from the uncertainty on the mineral concentration factors, $\Delta\alpha_i$, from:

$$\Delta\epsilon_i = \sqrt{\sum_{k=1}^{i}(\frac{\partial\epsilon_i}{\partial\alpha_k})^2\Delta\alpha_k^2} \tag{26}$$

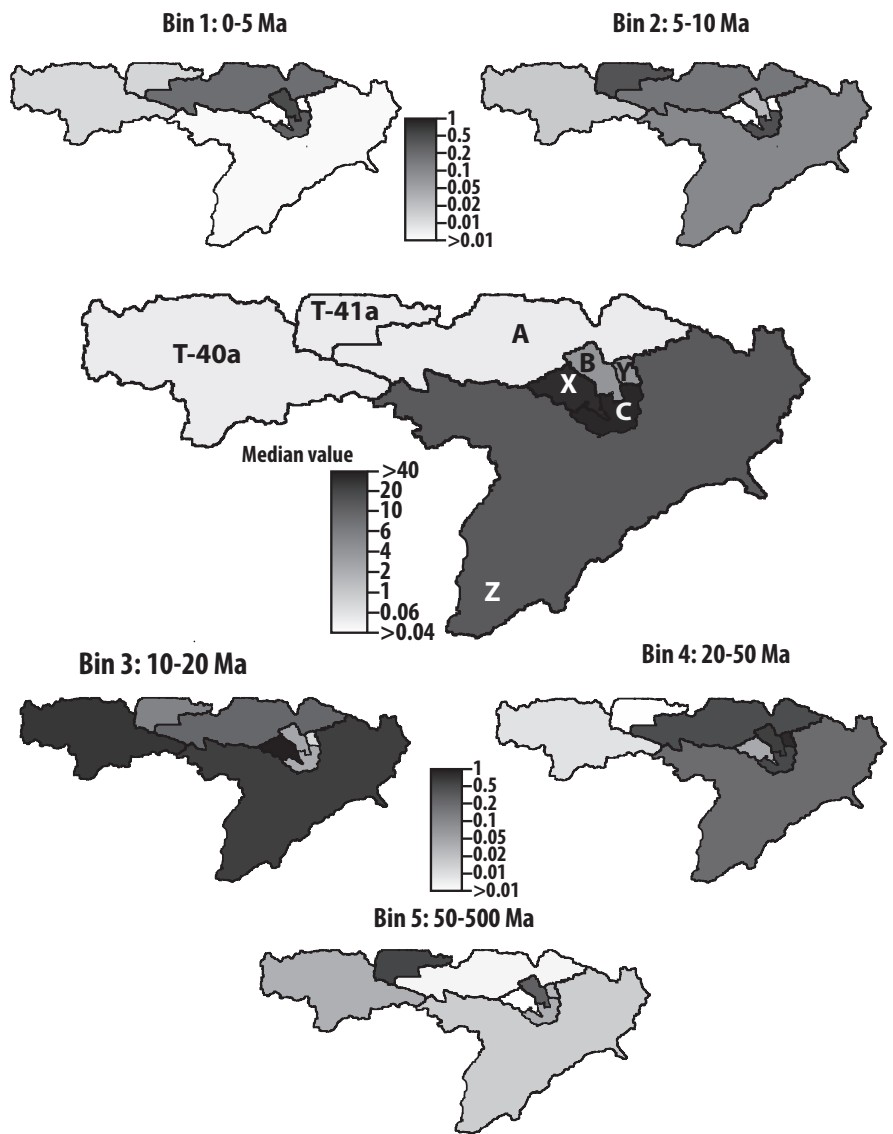

**Figure 9.** Maps of predicted median erosion rates (central panel) and relative surface age concentrations from the Muscovite detrital data from Eastern Himalaya. See Figures (8) and (8) for full distributions and data.

where:

$$\frac{\partial \epsilon_i}{\partial \alpha_k} = \left\{ \begin{array}{ll} 0 & \text{if } k > i \\ \dfrac{\epsilon_i}{\alpha_i} & \text{if } k = i \\ \dfrac{\delta_i^m}{F_i}\left(A_k \epsilon_k + \sum_{j=1}^{i-1} F_j \dfrac{\partial \epsilon_j}{\partial \alpha_k}\right) & \text{if } k < i \end{array} \right\} \tag{27}$$

The results are shown in Figure 10 as a plot of the ratio between the relative uncertainty in estimates of erosion rate $\Delta \epsilon_i / \epsilon_i$ and the relative uncertainty in mineral concentration factors, $\Delta \alpha_i / \alpha_i$, for the six stations located along the main river trunk. We see that the relative uncertainty in erosion rate is approximately proportional to the relative uncertainty in mineral concentration factor (i.e. all values are close to 1) and that there is only a minor downstream propagation of the uncertainty . This is also a simple consequence of the incremental nature of our algorithm, as explained by Equation 11.

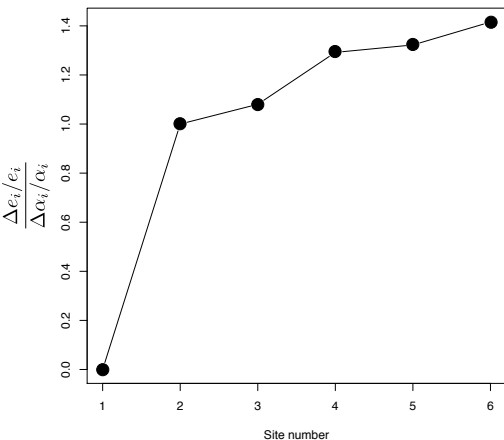

**Figure 10.** Relative uncertainty in erosion rate scaled by the relative uncertainty in mineral concentration factor for the estimates obtained at each of the six sites along the main river trunk. The first site has a fixed relative erosion rate (=1) and therefore no uncertainty.

## 7 Ways in which the method could be improved

As described in this paper, our methods relies on the existence of age clusters (or bins) that can be found in the age distributions collected at various sites along a river. The method could be generalized by constructing kernel density estimates of the distribution of ages at each sites. These could be used to estimate the minimum contribution factors, $\delta_i^m$, by applying condition (16) for the complete range of ages, not just the discrete values obtained by binning. We have found, however, that the solution we obtain in this way is strongly dependent on the choice made for the kernel and further investigation of this issue is required. Alternatively, estimates of cumulative density functions could be built directly from the data and, in turn, used to impose

condition (16) and estimate the minimum contribution factors, $\delta_i^m$. However the limited number of grains at each site makes the comparison between two successive CDFs rather inaccurate. The accuracy of this approach can be tested by increasing the number of bins in our model such that $N$ becomes similar to the average number of grains in any dataset. This leads to an overestimation of the minimum contribution factors. Clearly more work is required to investigate better or alternative ways to compare two successive age distributions.

## 8  Conclusions

We have developed a simple method to extract spatially variable erosion rates and surface age distributions from detrital cooling-age datasets from modern river sands. The method is based on what we believe are the simplest assumptions necessary to interpret such data and does not rely on a-priori knowledge of the distribution of ages in surrounding catchments. In describing the method we have demonstrated that it is suited to extract two seemingly sources of information pertaining to the spatial distribution of erosion rate along the river. The first comes from using the age distributions as fingerprints characterizing the areas between two successive sites where detrital samples were collected. This allows us to predict first-order estimates of the relative erosion rate between these areas and the distribution of ages in surficial rocks in each area. These estimates of age distributions can be used as a second independent information on the past and present erosion rate in each area.

By applying the method to an existing dataset from the eastern Himalaya, we show that the method provides estimates of present-day erosion rate patterns in the area that is consistent with previous, independent estimates, potentially evidencing that the fast present-day erosion rates in some parts of the study area are relatively young. We stress, however, that our method can only provide reliable estimates of erosion rate when the age distributions observed at two successive sites are different.

Importantly, the method is limited to providing the spatial distribution of erosion rate; independent information is necessary to transform those into absolute estimates of erosion rate.

## 9  Appendix

From the definition of the relative contributions, $\delta_i$:

$$\delta_i = \frac{\rho_i}{\sum_{j=1}^{i-1} \rho_j} \quad \text{for } i = 2, \cdots, M \tag{28}$$

where:

$$\rho_i = \frac{F_i \epsilon_i}{F_1 \epsilon_1} \tag{29}$$

we can write:

$$F_i \epsilon_i = \delta_i \sum_{j=1}^{i-1} F_j \epsilon_j = \delta_i \Big( \sum_{j=1}^{i-2} F_j \epsilon_j + F_{i-1} \epsilon_{i-1} \Big) = \delta_i \Big( \sum_{j=1}^{i-2} F_j \epsilon_j + \delta_{i-1} \sum_{j=1}^{i-2} F_j \epsilon_j \Big) = \delta_i (1 + \delta_{i-1}) \sum_{j=1}^{i-2} F_j \epsilon_j \tag{30}$$

By performing this operation $i - 2$ times and arbitrarily setting $\delta_1 = 0$, we obtain:

$$F_i \epsilon_i = \delta_i \prod_{j=1}^{i-1} (1 + \delta_j) F_1 \epsilon_1 \quad \text{for } i = 2, \cdots, M \tag{31}$$

*Code and data availability.* We provide a simple implementation of the method in python within a Jupyter Notebook that includes the data used in this paper for illustration purposes.

5 *Competing interests.* The authors declare that they have no conflict of interest.

*Acknowledgements.* The work leading to the results presented here was supported by the People Programme (Marie Curie Actions) of the European Union's Seventh Framework Programme FP7/People/2012/ITN, Grant agreement number 316966.

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
