# Peer review of "Extracting information on the spatial variability in erosion rate stored in detrital cooling age distributions in river sands"

_Earth Surface Dynamics, 2017_

## Referee Comment (RC1) · M. G. Malusa' (Referee) · 10 Aug 2017

**INTRODUCTION**

This section needs some improvements and reference to previous work, in order to properly emphasize the complexities of the thermochronologic record and the main assumptions of the detrital thermochronology approach (see below):

1) "Thermochronometric methods provide us with estimates of the cooling age of a rock, i.e. the time in the past when the rock cooled through a so-called closure temperature (Dodson, 1973), which varies between systems and minerals."

The concept of a closure temperature and a cooling age only applies in the case where rocks are cooling monotonically from high to low temperature (e.g., Dodson 1973; Villa 1998). For example, if a rock cools rapidly into the partial retention zone and is resident therein for a period of time before cooling again, its thermochronologic age cannot be recognized as a "cooling age". It is a common assumption in detrital thermochronology studies that all ages represent cooling ages, but this is not necessarily the case. This assumption should be properly underlined in the revised main text.

2) "One of the main geological processes through which rocks experience cooling is exhumation towards the cold, quasi-isothermal surface (Brown, 1991)."

I would underline here that the thermal reference frame relevant for isotopic closure is generally dynamic, which makes the interpretations of thermochronologic ages even more challenging, especially in detrital thermochronology.

3) "Young ages are commonly interpreted to indicate rapid exhumation and old ages should correspond to slow exhumation."

Old ages can also reflect denudation of shallow crustal levels that lay above the isothermal surface corresponding to the closure temperature of the thermochronologic system under consideration (e.g., Rahl et al. 2007).

4) "Cooling ages can also record more discrete cooling events such as the nearby emplacement of hot intrusions (Gleadow and Brooks, 1979) or the rapid relaxation of isotherms at the end of an episode of rapid erosion (Braun, 2016)."

A similar interpretation as Braun (2016) was also proposed for the European Southern Alps by Zanchetta et al. 2015 - Lithosphere. Cooling ages can also record thermal relaxation during the rifting to drifting transition (Malusà et al. 2016a - Gondwana Research), or mineral crystallization that has occurred at shallow crustal depth above the closure temperature isothermal surface (e.g., Malusà et al. 2011 - EPSL).

5) "Datasets are now routinely assembled by collecting and dating a large number of

mineral grains from a sand sample collected at a given location in a river draining an actively eroding area. Such detrital thermochronology datasets provide a proxy for the distribution of surface rock ages in a given catchment (Bernet et al., 2004; Brandon, 1992)"

This only applies in case of uniform mineral fertility in eroded rocks (Malusà et al. 2016b - Gondwana Research).

6) "By repeating this operation at different sites along a river stream, one obtains redundant information that can be used to document more precisely the spatial variability of in-situ thermochronological ages in a river catchment (Bernet et al., 2004; Brewer et al., 2006)."

The detrital thermochronologic record reflects both the thermochronologic complexities of eroded bedrock, and the bias acquired during erosion, transport and deposition (e.g., hydraulic sorting and mineral fertility bias, see Malusà et al. 2013 - Chemical Geology; Malusà et al. 2016b - Gondwana Research). All of these complexities and potential sources of bias should be properly taken into account and mentioned in the revised main text.

7) "However, these methods have not taken advantage of the fact that detrital age distributions contain two separate pieces of information concerning the spatial patterns of present and past rates of erosion. The first piece of information comes from the ages themselves: catchments or sub-catchments where the proportion of grains with young ages dominates are likely to experience rapid exhumation today or in the recent past; whereas catchments or sub-catchments where the proportion of grains with old ages dominates are more likely to have experienced rapid erosion in a more distant past."

This is not novel. The dual information (long-term vs short-term erosion/exhumation) provided by detrital thermochronology datasets was first discussed by Malusà et al. 2009 (Geol Soc London Spec Publ) under the assumption of constant mineral fertility in the eroding sources. This topic was further developed by Resentini and Malusà

2012 (Geol Soc Spec Papers) and Malusà et al. 2016b (Gondwana Research), taking into account the dishomogeneous mineral fertility in the source rocks. All these papers should be quoted in the revised manuscript.

THE METHOD

8) "In each Area i, we will assume that alpha i is the relative abundance of the mineral used to estimate the age distribution in rocks being eroded from the surface. We take the convention that 0 < alpha i < 1, with alpha i = 1 corresponding to an area i with surface rocks that contain the mineral in abundance (for example granite for muscovite) and alpha i = 0 corresponding to an area i with surface rocks that do not contain the mineral (for example carbonates for muscovite). If, for example, the area is made of 60% granite and 40% carbonates, and we have measured ages using a mineral that is abundant in granites (like muscovite) but absent in carbonates, then alpha = 0.6."

Alpha just provides a rough estimate of the mineral fertility bias. Malusà et al (2016b) demonstrated that major mineral fertility variations can be observed even in tectonic units with similar lithology, and showed that the relationships between bedrock geology and mineral fertility are complex and hardly predictable. They depend not only on lithology, but more in general on the whole magmatic, sedimentary or metamorphic evolution of eroded rocks. Careful approaches to mineral fertility measuremens are consequently required (see Malusà et al. 2016b - Gondwana Research). I think that this issue should be discussed in more detail in the revised manuscript.

APPLICATIONS TO DETRITAL AGE DISTRIBUTIONS

9) "Table 2" The lithological factor shown in Table 2 is very similar for different catchments. Is this correct? Was the mineral fertility measured accurately? Expected mineral fertility variations in Alpine-type orogenic belts should be on the order of 10e2-10e3 (see, e.g., Malusà et al. 2016b - Gondwana Research).

10) "Interestingly, there is a good correspondence between present-day erosion rate

and where the youngest ages are being generated (compare upper left panel showing relative concentration of youngest age bin, to central panel showing predicted present-day erosion rate), with the notable exception of the most downstream catchment (Z). In other words, where the mixing analysis predicts high erosion rate to account for a substantial change in the age distribution between two adjacent catchments, is also where it predicts the highest concentration of young ages in the surface rocks."

The short-term erosion rates calculated by Braun et al. are strongly influenced by the mineral fertility bias. Without an accurate measurement of mineral fertility and a proper consideration of hydraulic sorting effects, the comparison between long-term and short-term erosion rates performed here is rather weak.
* * *
References cited

Malusà, M. G., Zattin, M., Andò, S., Garzanti, E., & Vezzoli, G. (2009). Focused erosion in the Alps constrained by fission-track ages on detrital apatites. Geological Society, London, Special Publications, 324(1), 141-152.

Malusà, M. G., Villa, I. M., Vezzoli, G., & Garzanti, E. (2011). Detrital geochronology of unroofing magmatic complexes and the slow erosion of Oligocene volcanoes in the Alps. Earth and Planetary Science Letters, 301(1), 324-336.

Malusà, M. G., Carter, A., Limoncelli, M., Villa, I. M., & Garzanti, E. (2013). Bias in detrital zircon geochronology and thermochronometry. Chemical Geology, 359, 90-107.

Malusà, M. G., Danišík, M., & Kuhlemann, J. (2016a). Tracking the Adriatic-slab travel beneath the Tethyan margin of Corsica–Sardinia by low-temperature thermochronometry. Gondwana Research, 31, 135-149.

Malusà, M. G., Resentini, A., & Garzanti, E. (2016b). Hydraulic sorting and mineral fertility bias in detrital geochronology. Gondwana Research, 31, 1-19.

Rahl, J. M., Ehlers, T. A., & van der Pluijm, B. A. (2007). Quantifying transient erosion of orogens with detrital thermochronology from syntectonic basin deposits. Earth and Planetary Science Letters, 256(1), 147-161.

Resentini, A., & Malusà, M. G. (2012). Sediment budgets by detrital apatite fission-track dating (Rivers Dora Baltea and Arc, Western Alps). Geological Society of America Special Papers, 487, 125-140.

Villa, I. M. (1998). Isotopic closure. Terra Nova-Oxford, 10(1), 42-47.

Zanchetta, S., Malusà, M. G., & Zanchi, A. (2015). Precollisional development and Cenozoic evolution of the Southalpine retrobelt (European Alps). Lithosphere, 7(6), 662-681.

---

## Editor Comment (EC1) · S. Castelltort (Editor) · 28 Oct 2017

I first would like to apologize to the authors for the very long time it has taken to process their paper. As associate editors our job is to find reviewers and try to get their reviews in due time. In this case, it has taken an unacceptable time before the decision had to be made to take over the review from an uber-uber-late reviewer.

I find the manuscript basically acceptable as is. It is of fundamental interest and potentially of broad applicability. I am far from understanding the math details underlying the approach presented, but I trust the authors and future users of the method to deal with this if needed.

[Figure]

I have a few comments/questions that the authors are free to consider.

"The first piece of information comes from the ages themselves: catchments or sub-catchments where the proportion of grains with young ages dominates are likely to experience rapid exhumation today or in the recent past; whereas catchments or sub-catchments where the proportion of grains with old ages dominates are more likely to have experienced rapid erosion in a more distant past."

=> Why would "a catchment with old ages" be interpreted as representing an area of rapid erosion in the past. I thought old ages meant slow erosion. Are you only saying that pulses of rapid erosion can't be resolved by thermochronometric method when ages are old? Hence only catchment with young ages would be able to decipher rapid erosion. Isn't there some bias here?

"For this, ages can be regarded as passive markers (or colors) that inform us on the proportion in which the mixing takes place today, which is directly proportional to the present-day erosion rate."

=> what is the relation that determines a direct proportionality link between the "mixing of passive markers in a river" and "present-day erosion rate". Is this obvious or are there any references to back this up?

"we have devised a simple method that, unlike many others such as that of Brewer et al. (2006), is only dependent on the raw, binned age data." I guess I understand that here you're bypassing the need to model individual age data into cooling rates through assumptions of geothermal gradients etc... If that's correct I must say that for a non-specialist it would be great to have a bit more material here on the assumptions that are made and not made. Also, are the ages really "raw" or do they come with uncertainty/standard error on them? And if yes, what are the uncertainties/SE on "raw" ages.

In the abstract it is said "We show that detrital age distributions contain dual information

about present-day erosion rate" but in the text it is more an assumption than a demonstration. And I also failed to see how the results obtained are confronted with existing constraints on erosion rates in this area.

4. Uncertainty estimates. Since the distribution are not normal, does it makes sense to use the standard deviation around the mode? Also, wouldn't it be possible to perform a standard error propagation that would include the standard errors on ages?

Conclusion: thanks to the authors for submitting their work to eSurf and apologies again for the slow process.

---

## Referee Comment (RC2) · M. Brandon (Referee) · 30 Oct 2017

Mark Brandon, Yale University October 30, 2017

Review of "Extracting information on the spatial variability in erosion rate stored in detrital cooling age distributions in river sands", by Jean Braun, Lorenzo Gemignani, and Peter van der Beek For consideration for Earth Surface Dynamics.

Recommendation This paper provides an approach for decomposing erosion rates from detrital cooling ages collected from multiple tributaries. The approach is innovative but the quantitative formulation is difficult to follow and the implementation has

major problems that undermine confidence in the results. To be blunt, I have no idea if the proposed formulations give the "right answer". I highlight these problems in my general comments below, and I follow with some specific comments. The paper is not suitable for full publication in its present form, but I think some careful revisions could transform the paper into an important contribution.

General Comments (see Specific Comments below for more details) 1) The paper starts out with a clever idea, to use detrital cooling ages from multiple tributaries to resolve relative modern erosion rates for each of the tributaries. The starting point is excellent.

2) The paper claims to be the first to use detrital thermochronometric data as a tracer for estimating modern erosion rates. This tracer approach has already been introduced by McPhillips and Brandon (2010) and Ehlers et al. (2015). The specific contribution here, using detrital thermochronology as a tracer from multiple nested catchments, is a new and important.

3) There is actually a lot of literature on the formulation and solution of mixing models. I would expect a brief summary of that literature, and also some discussion about advantages and disadvantages of previous methods and the new method presented in the paper. One analysis that I like is in Menke (2013, p. 10-11, 189-199).

4) The main contribution of this paper is a computation procedure that uses observed detrital cooling ages collected from tributary catchments and along the trunk stream of a large drainage to estimate average relative erosion rates for each of the tributary catchments. In other words, the estimation involves inverting the data to find best-fit solutions (expectations and confidence intervals) for the relative erosion rates. Inverse estimation is a well-established field and it makes sense to structure the problem in terms of this methodology. To do so requires a clear definition of the model equation and error function, and the determination of a computation method to optimize the unknown parameters relative to the observed data, using either least squares or likelihood. The estimation suggested in the paper provides no tie to statistical or inverse methodology, so it is difficult to know if the estimates will be correct.

5) The paper lacks any testing of the estimation method. The usual approach is to design a synthetic data set with noise, and use that to see if the estimation method recovers the parameters used to generate the synthetic data set. A successful test would show that as the size of the synthetic observed data is increased, the parameter estimates would asymptotically approach the "true" parameter values used to generate the synthetic dataset. I encourage this kind of test to be added to the paper.

6) I don't know why, but the authors decide that they can estimate the best-fit result and the uncertainties using a Monte Carlo simulation. They refer to this simulation as a "boot strap" estimation of uncertainties, but that is incorrect (see specific comment #4 below). In fact, they are using this simulation to estimate both the expectations and the uncertainties for the parameters. They note that they prefer the modes, and not the means, of the Monte Carlo distributions as estimators for the relative erosion rates. I understand their preference in that the Monte Carlo distributions are asymmetric, but they provide no evidence to show that the modes or the means work at all. In the end, it would make sense to solve the inverse problem directly, rather than rely on Monte Carlo distributions. Note that the bootstrap method is very useful non-parameteric method for estimating uncertainties. For the problem here, it probably makes sense to estimate bootstrap confidence intervals (see Carpenter and Bithell, 2000 for details), which require no assumptions about the shape of the bootstrap distribution.

7) There is no discussion of the structure of this estimation problem. Is it overdetermined, underdetermined, or mix determined? One is left to wonder if the constraints (eqs. 15, 16) are handled in a way that is consistent and unbiased with respect to the estimation problem. What is the structure of the errors, and how are the errors accounted for in the estimation algorithm? There is a vagueness about the estimated quantities, whether they are absolute or relative erosion rates. This point should be stated upfront and maintained in consistent way throughout the paper.
8) It is not clear what quantities are being estimated. In the formulation, it would seem that $C_{k,i}$ are the primary parameters to be estimated (section 2.4), and the relative erosion rates are derived from these parameters. The values for $C_{k,i}$ are bounded to the range [0,1], which means that their range is truncated on both sides. Constraints are introduced in the formulation (eqs. 15, 16) but there is no assurance that this strategy will give the right answer. In statistics, the well established approach is to remove the truncations by transforming the parameters to a new scale. The logit transform is used for parameters that are bound to [0, 1], where logit(x) = ln(x /(1-x). A positivity constraint for erosion rates can be introduced by a log transform. These strategies commonly result in symmetric Gaussian-like distributions for the parameters, which means that the best-fit solution and confidence intervals are typically well defined. The authors have the view that it is somehow better to fit "raw binned age data" (p. 3, line 10), rather than a probability density function. The binned data are not "raw" in that they are smoothed by the box function used for the binning. The topic of kernel density estimation (KDE) was first established in the mid 1950's has been well defined since about the mid-1980's. What is clear is that the box function used in estimating a histogram is just one type of kernel function. A Gaussian is a much better kernel function for estimating a density distribution. It would make no difference if one used a histogram versus a density distribution for this problem. Silverman (1986) provides a general review of estimating density functions, Brandon (1996) show an extension of the KDE method for use with grain ages with specified standard errors, and McPhillips and Brandon (2010) show how to combine estimated probability density functions to get a relative density function for tracer thermochronology. All of this approach is completely consistent with the formulations proposed in this manuscript. Note that Vermeesch's (2012) paper on grain age distributions provides nothing new to this issue of density estimation.

9) The authors have an application paper, Gemignani et al, 2017, which was published in August in Tectonics. The paper considered here makes no mention of this paper. It is important to provide some explanation of how that paper relates to this contribution.

Specific Comments 1) p. 2, lines 27-28: The paper states that previous publications have not taken advantage of the ability of thermochronologic data to resolve both past and present erosion rates. In fact, McPhillips and Brandon (2010) was entirely devoted to showing how thermochronology can be used as a tracer to estimate modern erosion rates. Ehlers et al. (2015) also has a similar application.

2) p. 3, lines 9-10, 29-30: Not clear why bins are better why to represent the density of the data. The authors imply that the bins can be tuned to an 'event of given "age"', but there is no explanation about why this capability is important or even desired. In addition, there are the usual questions about histograms: How many bins should be used?, How wide should the bins be?, etc.

3) p. 6, Incremental Formulation: This section provides another solution for the estimation problem. It would help if there were some explanation about why a second approach is needed.

4) p. 9, line 12: The numerical estimation is described here as a bootstrap, but the method used is not the bootstrap (Efron and Tibshirani, 1986), but rather an ad-hoc procedure. I am puzzled here because the bootstrap calculation is very simple (replicate data sets produced by random sampling with replacement of the original data set), and it has well defined properties for estimation of uncertainties. In contrast, I have no idea if the ad-hoc procedure used here (randomly removing 25% of the data) is able to provide reliable estimates of uncertainties.

5) p. 9, line 27: It would help to explain here why the closure temperature for Ar muscovite is cited here, given that this information is not used in the paper.

6) p. 12, fig. 4: The horizontal axes have no tic values or axis labels, and the vertical axes are also unlabeled.

7) p. 13, lines 5-9: The estimation method seems to be rather unstable.

8) p. 14, figure 5: This figure is hard to understand. It is my guess that the gray scale

represents, not the estimated erosion rate, but rather the estimated relative erosion rate. Is that correct?

Cited References Brandon, M.T., 1996. Probability density plot for fission-track grain-age samples. Radiation Measurements 26, 663–676.

Carpenter, J., Bithell, J., 2000. Bootstrap confidence intervals: when, which, what? A practical guide for medical statisticians. Stat Med 19, 1141–1164.

Efron, B., Tibshirani, R., 1986. Bootstrap methods for standard errors, confidence intervals, and other measures of statistical accuracy. Stat Sci 1, 54–75. doi:10.1214/ss/1177013815

Ehlers, Todd A., Annika Szameitat, Eva Enkelmann, Brian J. Yanites, and Glenn J. Woodsworth. "Identifying spatial variations in glacial catchment erosion with detrital thermochronology." Journal of Geophysical Research: Earth Surface 120, no. 6 (2015): 1023-1039.

Gemignani, Lorenzo, Xilin Sun, J. Braun, T. D. Gerve, and Jan Robert Wijbrans. "A new detrital mica 40Ar/39Ar dating approach for provenance and exhumation of the Eastern Alps." Tectonics 36, no. 8 (2017): 1521-1537.

McPhillips, D. and Brandon, M.T., 2010. Using tracer thermochronology to measure modern relief change in the Sierra Nevada, California. Earth and Planetary Science Letters, 296(3), pp.373-383.

Menke, William. Geophysical data analysis: discrete inverse theory: MATLAB edition. Vol. 45. Academic press, 2012.

Silverman, B.W., 1986. Density estimation for statistics and data analysis. Chapman & Hall/CRC.

Vermeesch, P., 2012. On the visualisation of detrital age distributions. Chemical Geology 312-313, 190–194. doi:10.1016/j.chemgeo.2012.04.021

**ESurfD**

Interactive
comment

---

## Author Comment (AC1) · 8 Dec 2017

We would like to thank the Reviewers and the Associate Editor for their very useful and constructive comments concerning our work. It is clear that, in preparing a revised version, we need to improve our referencing our previous work on the subject, better warn the reader that a proper correction for mineral fertility bias needs to be applied and better describe the motivation and the assumptions on which the method is based and has been developed. We also need to improve the description of the method itself.

[Figure]

**1   Reviewer 1's comments**

This section needs some improvements and reference to previous work, in order to properly emphasize the complexities of the thermochronologic record and the main assumptions of the detrital thermochronology approach (see below):

1.  *"Thermochronometric methods provide us with estimates of the cooling age of a rock, i.e. the time in the past when the rock cooled through a so-called closure temperature (Dodson, 1973), which varies between systems and minerals." The concept of a closure temperature and a cooling age only applies in the case where rocks are cooling monotonically from high to low temperature (e.g., Dodson 1973; Villa 1998). For example, if a rock cools rapidly into the partial retention zone and is resident therein for a period of time before cooling again, its thermochronologic age cannot be recognized as a cooling age. It is a common assumption in detrital thermochronology studies that all ages represent cooling ages, but this is not necessarily the case. This assumption should be properly underlined in the revised main text.*

    We will change the sentence by removing the reference to the closure temperature or better explain it to avoid unnecessary confusion, although we dont fully understand what the referee means by a thermochronometric age cannot be recognized as a cooling age .

2.  *"One of the main geological processes through which rocks experience cooling is exhumation towards the cold, quasi-isothermal surface (Brown, 1991)." I would underline here that the thermal reference frame relevant for isotopic closure is generally dynamic, which makes the interpretations of thermochronologic ages even more challenging, especially in detrital thermochronology.*

    We dont fully understand the comment by the reviewer, i.e. what he means by "general dynamic".

3. *"Young ages are commonly interpreted to indicate rapid exhumation and old ages should correspond to slow exhumation." Old ages can also reflect denudation of shallow crustal levels that lay above the isothermal surface corresponding to the closure temperature of the thermochronologic system under consideration (e.g., Rahl et al. 2007).*

   We will be more careful in rewriting this sentence, but to get a younger rock next to an older rock (in a thermochronological sense), one needs to have exhumed one more than the other, and therefore one faster than the other, everything else being considered.

4. *"Cooling ages can also record more discrete cooling events such as the nearby emplacement of hot intrusions (Gleadow and Brooks, 1979) or the rapid relaxation of isotherms at the end of an episode of rapid erosion (Braun, 2016)." A similar interpretation as Braun (2016) was also proposed for the European Southern Alps by Zanchetta et al. 2015 - Lithosphere. Cooling ages can also record thermal relaxation during the rifting to drifting transition (Malusà et al. 2016a - Gondwana Research), or mineral crystallization that has occurred at shallow crustal depth above the closure temperature isothermal surface (e.g., Malusa et al. 2011 - EPSL).*

   We will cite appropriate references but do not agree that Malusa et al 2011 EPSL is the best citation for the resetting of ages by mineral crystallization; furthermore the rapid cooling caused by relaxing isotherms was already mentioned by Kellet et al 2013 Tectonics, which is cited in Braun (2016); we feel however that the citation to Braun (2016) is more appropriate as it focuses on this effect and provides the first comprehensive quantification of it.

5. *"Datasets are now routinely assembled by collecting and dating a large number of mineral grains from a sand sample collected at a given location in a river draining an actively eroding area. Such detrital thermochronology datasets provide*

**ESurfD**
*a proxy for the distribution of surface rock ages in a given catchment (Bernet et al., 2004; Brandon, 1992)"* This only applies in case of uniform mineral fertility in eroded rocks (Malusà et al. 2016b - Gondwana Research).

We don't fully understand the comment from the reviewer. Even if mineral fertility needs to be taken into account, the ages are a still a proxy for the distribution of surface rock ages in a given catchment (unless one considers that sand grain can "jump" from one catchment to the other)

6. *"By repeating this operation at different sites along a river stream, one obtains redundant information that can be used to document more precisely the spatial variability of in-situ thermochronological ages in a river catchment (Bernet et al., 2004; Brewer et al., 2006)."* The detrital thermochronologic record reflects both the thermochronologic complexities of eroded bedrock, and the bias acquired during erosion, transport and deposition (e.g., hydraulic sorting and mineral fertility bias, see Malusà et al. 2013 - Chemical Geology; Malusà et al. 2016b - Gondwana Research). All of these complexities and potential sources of bias should be properly taken into account and mentioned in the revised main text.

We have attempted to include the so-called fertility bias into the model by introducing what we call the "alpha" parameter. We will include a short paragraph that explains better what the alpha parameter means, but our paper is not the place to review (and cite) all the literature on the subject. If there are reliable means to estimate the fertility bias, anyone using our proposed method should, of course, use it. But it is not the purpose of our paper to enter into this discussion.

7. *"However, these methods have not taken advantage of the fact that detrital age distributions contain two separate pieces of information concerning the spatial patterns of present and past rates of erosion. The first piece of information comes from the ages themselves: catchments or sub-catchments where the proportion of grains with young ages dominates are likely to experience rapid exhumation*

*today or in the recent past; whereas catchments or sub-catchments where the proportion of grains with old ages dominates are more likely to have experienced rapid erosion in a more distant past."* This is not novel. The dual information (long-term vs short-term erosion/exhumation) provided by detrital thermochronology datasets was first discussed by Malusà et al. 2009 (Geol Soc London Spec Publ) under the assumption of constant mineral fertility in the eroding sources. This topic was further developed by Resentini and Malusà 2012 (Geol Soc Spec Papers) and Malusà et al. 2016b (Gondwana Research), taking into account the dishomogeneous mineral fertility in the source rocks. All these papers should be quoted in the revised manuscript.

We will cite these references in the introduction.

8. *"In each Area i, we will assume that alpha i is the relative abundance of the mineral used to estimate the age distribution in rocks being eroded from the surface. We take the convention that 0 < alpha i < 1, with alpha i = 1 corresponding to an area i with surface rocks that contain the mineral in abundance (for example granite for muscovite) and alpha i = 0 corresponding to an area i with surface rocks that do not contain the mineral (for example carbonates for muscovite). If, for example, the area is made of 60% granite and 40% carbonates, and we have measured ages using a mineral that is abundant in granites (like muscovite) but absent in carbonates, then alpha = 0.6."* Alpha just provides a rough estimate of the mineral fertility bias. Malusà et al (2016b) demonstrated that major mineral fertility variations can be observed even in tectonic units with similar lithology, and showed that the relationships between bedrock geology and mineral fertility are complex and hardly predictable. They depend not only on lithology, but more in general on the whole magmatic, sedimentary or metamorphic evolution of eroded rocks. Careful approaches to mineral fertility measurements are consequently required (see Malusà et al. 2016b - Gondwana Research). I think that this issue should be discussed in more detail in the revised manuscript.*

We will add a short paragraph on mineral fertility bias (see our response to comment 7 above).

9. *"Table 2" The lithological factor shown in Table 2 is very similar for different catchments. Is this correct? Was the mineral fertility measured accurately? Expected mineral fertility variations in Alpine-type orogenic belts should be on the order of 10e2-10e3 (see, e.g., Malusà et al. 2016b - Gondwana Research).*

We have used first-order approximations for the "alpha" (or fertility) parameter that are based on a simple interpretation of the distribution of rock type at the surface of the various catchments. As stated above, if more accurate methods exist to estimate this parameter (and the data exists to infer them) they should obviously be used.

10. *"Interestingly, there is a good correspondence between present-day erosion rate C4 and where the youngest ages are being generated (compare upper left panel showing relative concentration of youngest age bin, to central panel showing predicted present- day erosion rate), with the notable exception of the most downstream catchment (Z). In other words, where the mixing analysis predicts high erosion rate to account for a substantial change in the age distribution between two adjacent catchments, is also where it predicts the highest concentration of young ages in the surface rocks." The short-term erosion rates calculated by Braun et al. are strongly influenced by the mineral fertility bias. Without an accurate measurement of mineral fertility and a proper consideration of hydraulic sorting effects, the comparison between long-term and short-term erosion rates performed here is rather weak.*

We will state that an accurate measurement of mineral fertility is needed.

**2 Reviewer 2's comments**

*Review of "Extracting information on the spatial variability in erosion rate stored in detrital cooling age distributions in river sands", by Jean Braun, Lorenzo Gemignani, and Peter van der Beek For consideration for Earth Surface Dynamics. Recommendation This paper provides an approach for decomposing erosion rates from detrital cooling ages collected from multiple tributaries. The approach is inno- vative but the quantitative formulation is difficult to follow and the implementation has major problems that undermine confidence in the results. To be blunt, I have no idea if the proposed formulations give the "right answer". I highlight these problems in my general comments below, and I follow with some specific comments. The paper is not suitable for full publication in its present form, but I think some careful revisions could transform the paper into an important contribution. General Comments (see Specific Comments below for more details)*

1. *The paper starts out with a clever idea, to use detrital cooling ages from multiple tributaries to resolve relative modern erosion rates for each of the tributaries. The starting point is excellent.*

2. *The paper claims to be the first to use detrital thermochronometric data as a tracer for estimating modern erosion rates. This tracer approach has already been introduced by McPhillips and Brandon (2010) and Ehlers et al. (2015). The specific contribution here, using detrital thermochronology as a tracer from multiple nested catchments, is a new and important.*

3. *There is actually a lot of literature on the formulation and solution of mixing models. I would expect a brief summary of that literature, and also some discussion about advantages and disadvantages of previous methods and the new method presented in the paper. One analysis that I like is in Menke (2013, p. 10-11, 189-199).*
We will include relevant references

4. *The main contribution of this paper is a computation procedure that uses observed detrital cooling ages collected from tributary catchments and along the trunk stream of a large drainage to estimate average relative erosion rates for each of the tributary catchments. In other words, the estimation involves inverting the data to find best-fit solutions (expectations and confidence intervals) for the relative erosion rates. Inverse estimation is a well-established field and it makes sense to structure the problem in terms of this methodology. To do so requires a clear definition of the model equation and error function, and the determination of a computation method to optimize the unknown parameters relative to the observed data, using either least squares or likelihood. The estimation suggested in the paper provides no tie to statistical or inverse methodology, so it is difficult to know if the estimates will be correct.*

As seen from many fo the reviewer's comments, we will need to improve the presentation of our method, and also more clearly state its main objectives and assumptions.

5. *The paper lacks any testing of the estimation method. The usual approach is to design a synthetic data set with noise, and use that to see if the estimation method recovers the parameters used to generate the synthetic data set. A successful test would show that as the size of the synthetic observed data is increased, the parameter estimates would asymptotically approach the "true" parameter values used to generate the synthetic dataset. I encourage this kind of test to be added to the paper.*

We will include a demonstration of the accuracy and usefulness of the method using synthetic data. In particular we will highlight the importance of having different "age signatures" in different catchments to obtain good estimates of the relative erosion rates. We will also complete the last section of the manuscript to

show how error estimates on ages and "fertility" affect the erosion rate estimates.

6. *I don't know why, but the authors decide that they can estimate the best-fit result and the uncertainties using a Monte Carlo simulation. They refer to this simulation as a "boot strap" estimation of uncertainties, but that is incorrect (see specific comment 4 below). In fact, they are using this simulation to estimate both the expectations and the uncertainties for the parameters. They note that they prefer the modes, and not the means, of the Monte Carlo distributions as estimators for the relative erosion rates. I understand their preference in that the Monte Carlo distributions are asymmetric, but they provide no evidence to show that the modes or the means work at all. In the end, it would make sense to solve the inverse problem directly, rather than rely on Monte Carlo distributions. Note that the bootstrap method is very useful non-parameteric method for estimating uncertainties. For the problem here, it probably makes sense to estimate bootstrap confidence intervals (see Carpenter and Bithell, 2000 for details), which require no assumptions about the shape of the bootstrap distribution.*

   See our response to specific comment 4

7. *There is no discussion of the structure of this estimation problem. Is it overdetermined, underdetermined, or mix determined? One is left to wonder if the constraints (eqs. 15, 16) are handled in a way that is consistent and unbiased with respect to the estimation problem. What is the structure of the errors, and how are the errors accounted for in the estimation algorithm? There is a vagueness about the estimated quantities, whether they are absolute or relative erosion rates. This point should be stated upfront and maintained in consistent way throughout the paper.*

   The problem is underdetermined and requires an additional constraint; in our case, we search for the solution that minimizes relative changes in erosion rate between successive catchments (as already indicated in the text). We will improve our manuscript to make this point clearer.

8. *It is not clear what quantities are being estimated. In the formulation, it would seem that Ck,i are the primary parameters to be estimated (section 2.4), and the relative erosion rates are derived from these parameters. The values for Ck,i are bounded to the range [0,1], which means that their range is truncated on both sides. Constraints are introduced in the formulation (eqs. 15, 16) but there is no assurance that this strategy will give the right answer. In statistics, the well established approach is to remove the truncations by transforming the parameters to a new scale. The logit transform is used for parameters that are bound to [0, 1], where logit(x) = ln(x /(1-x). A positivity constraint for erosion rates can be introduced by a log transform. These strategies commonly result in symmetric Gaussian-like distributions for the parameters, which means that the best-fit solution and confidence intervals are typically well defined. The authors have the view that it is somehow better to fit "raw binned age data" (p. 3, line 10), rather than a probability density function. The binned data are not "raw" in that they are smoothed by the box function used for the binning. The topic of kernel density estimation (KDE) was first established in the mid 1950's has been well defined since about the mid- 1980's. What is clear is that the box function used in estimating a histogram is just one type of kernel function. A Gaussian is a much better kernel function for estimating a density distribution. It would make no difference if one used a histogram versus a density distribution for this problem. Silverman (1986) provides a general review of estimating density functions, Brandon (1996) show an extension of the KDE method for use with grain ages with specified standard errors, and McPhillips and Brandon (2010) show how to combine estimated probability density functions to get a relative density function for tracer thermochronology. All of this approach is completely consistent with the formulations proposed in this manuscript. Note that Vermeesch's (2012) paper on grain age distributions provides nothing new to this issue of density estimation.*

We will discuss the effect of using raw age data and will compare our strategy to one consisting in using reconstructed density functions. We will also consider the other suggestions made by the reviewer.

9. *The authors have an application paper, Gemignani et al, 2017, which was published in August in Tectonics. The paper considered here makes no mention of this paper. It is important to provide some explanation of how that paper relates to this contribution.*

We will give reference to the Tectonics paper.

Specific Comments

1. *p. 2, lines 27-28: The paper states that previous publications have not taken advantage of the ability of thermochronologic data to resolve both past and present erosion rates. In fact, McPhillips and Brandon (2010) was entirely devoted to showing how thermochronology can be used as a tracer to estimate modern erosion rates. Ehlers et al. (2015) also has a similar application.*

We will revise this sentence and add relevant references.

2. *p. 3, lines 9-10, 29-30: Not clear why bins are better why to represent the density of the data. The authors imply that the bins can be tuned to an event of given "age", but there is no explanation about why this capability is important or even desired. In addition, there are the usual questions about histograms: How many bins should be used?, How wide should the bins be?, etc.*

We will discuss this as stated above (comment 8)

3. *p. 6, Incremental Formulation: This section provides another solution for the estimation problem. It would help if there were some explanation about why a second approach is needed.*

It is the same solution/approach but expressed differently. We will make this point clearer.

4. *p. 9, line 12: The numerical estimation is described here as a bootstrap, but the method used is not the bootstrap (Efron and Tibshirani, 1986), but rather an ad-hoc procedure. I am puzzled here because the bootstrap calculation is very simple (replicate data sets produced by random sampling with replacement of the original data set), and it has well defined properties for estimation of uncertainties. In contrast, I have no idea if the ad-hoc procedure used here (randomly removing 25% of the data) is able to provide reliable estimates of uncertainties.*

   We agree that a bootstrap method should not be removing but replacing samples; we will fix this.

5. *p. 9, line 27: It would help to explain here why the closure temperature for Ar muscovite is cited here, given that this information is not used in the paper.*

   We will change this to make it more relevant.

6. *p. 12, fig. 4: The horizontal axes have no tic values or axis labels, and the vertical axes are also unlabeled.*

   We will improve this figure

7. *p. 13, lines 5-9: The estimation method seems to be rather unstable.*

   We think this clearly shows the reliability of our estimates.

8. *p. 14, figure 5: This figure is hard to understand. It is my guess that the gray scale represents, not the estimated erosion rate, but rather the estimated relative erosion rate. Is that correct?*

   Yes. We will fix this.
**3   Associate Editor's comments**

*I find the manuscript basically acceptable as is. It is of fundamental interest and potentially of broad applicability. I am far from understanding the math details underlying the approach presented, but I trust the authors and future users of the method to deal with this if needed. I have a few comments/questions that the authors are free to consider.*

- *"The first piece of information comes from the ages themselves: catchments or sub- catchments where the proportion of grains with young ages dominates are likely to experience rapid exhumation today or in the recent past; whereas catchments or sub- catchments where the proportion of grains with old ages dominates are more likely to have experienced rapid erosion in a more distant past." => Why would "a catchment with old ages" be interpreted as representing an area of rapid erosion in the past. I thought old ages meant slow erosion. Are you only saying that pulses of rapid erosion can't be resolved by thermochronometric method when ages are old? Hence only catchment with young ages would be able to decipher rapid erosion. Isn't there some bias here?*

  Of course, the editor is correct; we will change the sentence to reflect this point.

- *"For this, ages can be regarded as passive markers (or colors) that inform us on the proportion in which the mixing takes place today, which is directly proportional to the present-day erosion rate." => what is the relation that determines a direct proportionality link between the "mixing of passive markers in a river" and "present-day erosion rate". Is this obvious or are there any references to back this up?*

  Because faster present-day erosion rate will yield a proportionally larger sediment flux into the river, everything else being considered the same. We will adapt the sentence to avoid the confusion caused to the associate editor.

**ESurfD**
- *"we have devised a simple method that, unlike many others such as that of Brewer et al. (2006), is only dependent on the raw, binned age data." I guess I understand that here you're bypassing the need to model individual age data into cooling rates through assumptions of geothermal gradients etc. . . If that's correct I must say that for a non-specialist it would be great to have a bit more material here on the assumptions that are made and not made.*

  We will change the text to make this clearer

- *Also, are the ages really "raw" or do they come with uncertainty/standard error on them? And if yes, what are the uncertainties/SE on "raw" ages. In the abstract it is said "We show that detrital age distributions contain dual information about present-day erosion rate" but in the text it is more an assumption than a demonstration. And I also failed to see how the results obtained are confronted with existing constraints on erosion rates in this area.*

  We will add references to previous work that shows that present-day erosion rates are highest near the eastern syntax, as predicted by our simple model.

- *Uncertainty estimates. Since the distribution are not normal, does it makes sense to use the standard deviation around the mode? Also, wouldn't it be possible to perform a standard error propagation that would include the standard errors on ages?*

  We will provide a better estimate of error estimate by performing model simulations on synthetic data (see response to second reviewer).

*Conclusion: thanks to the authors for submitting their work to eSurf and apologies again for the slow process.*

---

## Editor Comment (EC2) · S. Castelltort (Editor) · 21 Dec 2017

After considerations of reviewers comments and authors' response I encourage Braun and colleagues to submit a revised version of their manuscript.

---

## Author Response (AR2)

"Extracting information on the spatial variability in erosion rate stored in detrital cooling age distributions in river sands" by Braun et al.

**Reviewer #1's comments**

This section needs some improvements and reference to previous work, in order to properly emphasize the complexities of the thermochronologic record and the main assumptions of the detrital thermochronology approach (see below):

1) "Thermochronometric methods provide us with estimates of the cooling age of a rock, i.e. the time in the past when the rock cooled through a so-called closure temperature (Dodson, 1973), which varies between systems and minerals."

The concept of a closure temperature and a cooling age only applies in the case where rocks are cooling monotonically from high to low temperature (e.g., Dodson 1973; Villa 1998). For example, if a rock cools rapidly into the partial retention zone and is resident therein for a period of time before cooling again, its thermochronologic age cannot be recognized as a "cooling age". It is a common assumption in detrital thermochronology studies that all ages represent cooling ages, but this is not necessarily the case. This assumption should be properly underlined in the revised main text.

**We have change the sentence.**

2) "One of the main geological processes through which rocks experience cooling is exhumation towards the cold, quasi-isothermal surface (Brown, 1991)."

I would underline here that the thermal reference frame relevant for isotopic closure is generally dynamic, which makes the interpretations of thermochronologic ages even more challenging, especially in detrital thermochronology.

**We don't fully understand the comment by the reviewer, i.e. what he means by "generally dynamic".**

3) "Young ages are commonly interpreted to indicate rapid exhumation and old ages should correspond to slow exhumation."

Old ages can also reflect denudation of shallow crustal levels that lay above the isothermal surface corresponding to the closure temperature of the thermochronologic system under consideration (e.g., Rahl et al. 2007).

**We have rewritten this sentence to make it clearer.**

4) "Cooling ages can also record more discrete cooling events such as the nearby emplacement of hot intrusions (Gleadow and Brooks, 1979) or the rapid relaxation of isotherms at the end of an episode of rapid erosion (Braun, 2016)."

A similar interpretation as Braun (2016) was also proposed for the European Southern Alps by Zanchetta et al. 2015 - Lithosphere. Cooling ages can also record thermal relaxation during the rifting to drifting transition (Malusà et al. 2016a - Gondwana Research), or mineral crystallization that has occurred at shallow crustal depth above the closure temperature isothermal surface (e.g., Malusà et al. 2011 - EPSL).

**We have added appropriate references.**

5) "Datasets are now routinely assembled by collecting and dating a large number of mineral grains from a sand sample collected at a given location in a river draining

an actively eroding area. Such detrital thermochronology datasets provide a proxy for the distribution of surface rock ages in a given catchment (Bernet et al., 2004; Brandon, 1992)"

This only applies in case of uniform mineral fertility in eroded rocks (Malusà et al. 2016b - Gondwana Research).

**We have rewritten this sentence and added a small discussion on the importance of estimating what we refer to as mineral surface concentration factors and the reviewer calls "fertility".**

6) "By repeating this operation at different sites along a river stream, one obtains redundant information that can be used to document more precisely the spatial variability of in-situ thermochronological ages in a river catchment (Bernet et al., 2004; Brewer et al., 2006)."

The detrital thermochronologic record reflects both the thermochronologic complexities of eroded bedrock, and the bias acquired during erosion, transport and deposition (e.g., hydraulic sorting and mineral fertility bias, see Malusà et al. 2013 - Chemical Geology; Malusà et al. 2016b - Gondwana Research). All of these complexities and potential sources of bias should be properly taken into account and mentioned in the revised main text.

**We have include a short paragraph on the importance of the fertility factor mentioned by the reviewer. We have also included a better explanation of what our alpha parameter means, but our paper is not the place to review (and cite) all the literature on the subject. If there are reliable means to estimate the fertility bias, anyone using our proposed method should, of course, use it. But it is not the purpose of our paper to enter into this discussion.**

7) "However, these methods have not taken advantage of the fact that detrital age distributions contain two separate pieces of information concerning the spatial patterns of present and past rates of erosion. The first piece of information comes from the ages themselves: catchments or sub-catchments where the proportion of grains with young ages dominates are likely to experience rapid exhumation today or in the recent past; whereas catchments or sub-catchments where the proportion of grains with old ages dominates are more likely to have experienced rapid erosion in a more distant past."

This is not novel. The dual information (long-term vs short-term erosion/exhumation) provided by detrital thermochronology datasets was first discussed by Malusà et al. 2009 (Geol Soc London Spec Publ) under the assumption of constant mineral fertility in the eroding sources. This topic was further developed by Resentini and Malusà 2012 (Geol Soc Spec Papers) and Malusà et al. 2016b (Gondwana Research), taking into account the dishomogeneous mineral fertility in the source rocks. All these papers should be quoted in the revised manuscript.

**We have cited the work by Resentini and Maulsa 2016b which uses a similar, yet different method. In their analysis they assume that they know to which peak(s) in the age distributions each of the subcatchment contributes to (similar to assuming the relative values of the Cik in our method; see Figure 8 in Resentini and Malusa 2016)**

8) "In each Area i, we will assume that alpha i is the relative abundance of the mineral used to estimate the age distribution in rocks being eroded from the

surface. We take the convention that $0 < \alpha_i < 1$, with $\alpha_i = 1$ corresponding to an area $i$ with surface rocks that contain the mineral in abundance (for example granite for muscovite) and $\alpha_i = 0$ corresponding to an area $i$ with surface rocks that do not contain the mineral (for example carbonates for muscovite). If, for example, the area is made of 60% granite and 40% carbonates, and we have measured ages using a mineral that is abundant in granites (like muscovite) but absent in carbonates, then $\alpha = 0.6$."

Alpha just provides a rough estimate of the mineral fertility bias. Malusà et al (2016b) demonstrated that major mineral fertility variations can be observed even in tectonic units with similar lithology, and showed that the relationships between bedrock geology and mineral fertility are complex and hardly predictable. They depend not only on lithology, but more in general on the whole magmatic, sedimentary or metamorphic evolution of eroded rocks. Careful approaches to mineral fertility measurements are consequently required (see Malusà et al. 2016b - Gondwana Research). I think that this issue should be discussed in more detail in the revised manuscript.

**As stated earlier, we have added a paragraph on mineral fertility bias .**

9) "Table 2" The lithological factor shown in Table 2 is very similar for different catchments. Is this correct? Was the mineral fertility measured accurately? Expected mineral fertility variations in Alpine-type orogenic belts should be on the order of 10e2-10e3 (see, e.g., Malusà et al. 2016b - Gondwana Research).

**In the application of our method to the Eastern Himalaya dataset, we have compared results obtained by using uniform values for the alpha parameters to those obtained by using first-order estimates of the alpha parameters derived from the geological map. Although the results are dependent on this choice, the most salient feature of the results (large increase in erosion rate near the syntaxis) is not affected.**

10) "Interestingly, there is a good correspondence between present-day erosion rate C4 and where the youngest ages are being generated (compare upper left panel showing relative concentration of youngest age bin, to central panel showing predicted present- day erosion rate), with the notable exception of the most downstream catchment (Z). In other words, where the mixing analysis predicts high erosion rate to account for a substantial change in the age distribution between two adjacent catchments, is also where it predicts the highest concentration of young ages in the surface rocks."

The short-term erosion rates calculated by Braun et al. are strongly influenced by the mineral fertility bias. Without an accurate measurement of mineral fertility and a proper consideration of hydraulic sorting effects, the comparison between long-term and short-term erosion rates performed here is rather weak.

**We have shown how the uncertainty on the alpha parameters influences the accuracy of the estimates of erosion rates.**

**Reviewer #2's comments**

Review of "Extracting information on the spatial variability in erosion rate stored in detrital cooling age distributions in river sands", by Jean Braun, Lorenzo Gemignani,

and Peter van der Beek For consideration for Earth Surface Dynamics.
Recommendation This paper provides an approach for decomposing erosion rates from detrital cooling ages collected from multiple tributaries. The approach is innovative but the quantitative formulation is difficult to follow and the implementation has major problems that undermine confidence in the results. To be blunt, I have no idea if the proposed formulations give the "right answer". I highlight these problems in my general comments below, and I follow with some specific comments. The paper is not suitable for full publication in its present form, but I think some careful revisions could transform the paper into an important contribution.
General Comments (see Specific Comments below for more details)

1) The paper starts out with a clever idea, to use detrital cooling ages from multiple tributaries to resolve relative modern erosion rates for each of the tributaries. The starting point is excellent.
2) The paper claims to be the first to use detrital thermochronometric data as a tracer for estimating modern erosion rates. This tracer approach has already been introduced by McPhillips and Brandon (2010) and Ehlers et al. (2015). The specific contribution here, using detrital thermochronology as a tracer from multiple nested catchments, is a new and important.
We have added a reference to McPhillips and Brandon (2010)
3) There is actually a lot of literature on the formulation and solution of mixing models. I would expect a brief summary of that literature, and also some discussion about advantages and disadvantages of previous methods and the new method presented in the paper. One analysis that I like is in Menke (2013, p. 10-11, 189-199).
**Our method assumes that ages represent distinct events that can be used to define age bins that are then used as tracers. To define those bins, one can construct kernel density estimates of the age distributions. A short sentence has been added to explain this. Alternatively, one could also use the ages distributions to construct estimates of cumulative density functions (CDFs) that could then be used to estimate the minimum required erosion rate to explain differences between two successive CDFs. After testing on synthetic datasets, we have tentatively concluded that such a method is, however, less accurate. This is reported in a new section entitled "Ways in which the method could be improved".**
4) The main contribution of this paper is a computation procedure that uses observed detrital cooling ages collected from tributary catchments and along the trunk stream of a large drainage to estimate average relative erosion rates for each of the tributary catchments. In other words, the estimation involves inverting the data to find best-fit solutions (expectations and confidence intervals) for the relative erosion rates. Inverse estimation is a well-established field and it makes sense to structure the problem in terms of this methodology. To do so requires a clear definition of the model equation and error function, and the determination of a computation method to optimize the unknown parameters relative to the observed data, using either least squares or likelihood. The estimation suggested in the paper provides no tie to statistical or inverse methodology, so it is difficult to know if the

estimates will be correct.

**We have lengthened and improved the description of the method and attempted to make it clearer and more rigorous.**

5) The paper lacks any testing of the estimation method. The usual approach is to design a synthetic data set with noise, and use that to see if the estimation method recovers the parameters used to generate the synthetic data set. A successful test would show that as the size of the synthetic observed data is increased, the parameter estimates would asymptotically approach the "true" parameter values used to generate the synthetic dataset. I encourage this kind of test to be added to the paper.

**We have include a demonstration of the accuracy and usefulness of the method using synthetic data. In particular we have highlighted the importance of having different "age signatures" in different catchments to obtain good estimates of the relative erosion rates.**

6) I don't know why, but the authors decide that they can estimate the best-fit result and the uncertainties using a Monte Carlo simulation. They refer to this simulation as a "boot strap" estimation of uncertainties, but that is incorrect (see specific comment #4 below). In fact, they are using this simulation to estimate both the expectations and the uncertainties for the parameters. They note that they prefer the modes, and not the means, of the Monte Carlo distributions as estimators for the relative erosion rates. I understand their preference in that the Monte Carlo distributions are asymmetric, but they provide no evidence to show that the modes or the means work at all. In the end, it would make sense to solve the inverse problem directly, rather than rely on Monte Carlo distributions. Note that the bootstrap method is very useful non-parameteric method for estimating uncertainties. For the problem here, it probably makes sense to estimate bootstrap confidence intervals (see Carpenter and Bithell, 2000 for details), which require no assumptions about the shape of the bootstrap distribution.

**We are now showing median values of the distributions obtained by bootstrapping, as well as lower and upper quartiles. This is based on comparing the model results with known erosion rates in the synthetic examples that suggests that median values are most appropriate.**

7) There is no discussion of the structure of this estimation problem. Is it overdetermined, underdetermined, or mix determined? One is left to wonder if the constraints (eqs. 15, 16) are handled in a way that is consistent and unbiased with respect to the estimation problem. What is the structure of the errors, and how are the errors accounted for in the estimation algorithm? There is a vagueness about the estimated quantities, whether they are absolute or relative erosion rates. This point should be stated upfront and maintained in consistent way throughout the paper.

**The problem is underdetermined. We have shown however, that we can obtained minimum values for the erosion rates in successive catchments to satisfy observed differences in age distributions between successive catchments. We have improved the manuscript to make this point clearer.**

8) It is not clear what quantities are being estimated. In the formulation, it would seem that $C_{k,i}$ are the primary parameters to be estimated (section 2.4), and the

relative erosion rates are derived from these parameters. The values for Ck,i are bounded to the range [0,1], which means that their range is truncated on both sides. Constraints are introduced in the formulation (eqs. 15, 16) but there is no assurance that this strategy will give the right answer. In statistics, the well established approach is to remove the truncations by transforming the parameters to a new scale. The logit transform is used for parameters that are bound to [0, 1], where logit(x) = ln(x /(1-x). A positivity constraint for erosion rates can be introduced by a log transform. These strategies commonly result in symmetric Gaussian-like distributions for the parameters, which means that the best-fit solution and confidence intervals are typically well defined. The authors have the view that it is somehow better to fit "raw binned age data" (p. 3, line 10), rather than a probability density function. The binned data are not "raw" in that they are smoothed by the box function used for the binning. The topic of kernel density estimation (KDE) was first established in the mid 1950's has been well defined since about the mid-1980's. What is clear is that the box function used in estimating a histogram is just one type of kernel function. A Gaussian is a much better kernel function for estimating a density distribution. It would make no difference if one used a histogram versus a density distribution for this problem. Silverman (1986) provides a general review of estimating density functions, Brandon (1996) show an extension of the KDE method for use with grain ages with specified standard errors, and McPhillips and Brandon (2010) show how to combine estimated probability density functions to get a relative density function for tracer thermochronology. All of this approach is completely consistent with the formulations proposed in this manuscript. Note that Vermeesch's (2012) paper on grain age distributions provides nothing new to this issue of density estimation.

**See response to point (3)**

9) The authors have an application paper, Gemignani et al, 2017, which was published in August in Tectonics. The paper considered here makes no mention of this paper. It is important to provide some explanation of how that paper relates to this contribution.

**We have given reference to the Tectonics paper in the introduction.**

Specific Comments

1) p. 2, lines 27-28: The paper states that previous publications have not taken advantage of the ability of thermochronologic data to resolve both past and present erosion rates. In fact, McPhillips and Brandon (2010) was entirely devoted to showing how thermochronology can be used as a tracer to estimate modern erosion rates. Ehlers et al. (2015) also has a similar application.

**We have added a reference to McPhillips and Brandon (2010).**

2) p. 3, lines 9-10, 29-30: Not clear why bins are better why to represent the density of the data. The authors imply that the bins can be tuned to an 'event of given "age"', but there is no explanation about why this capability is important or even desired. In addition, there are the usual questions about histograms: How many bins should be used?, How wide should the bins be?, etc.

**See response to point (3)**

3) p. 6, Incremental Formulation: This section provides another solution for the estimation problem. It would help if there were some explanation about why a

second approach is needed.

**It is the same solution/approach but expressed differently. We have made this point clearer.**

4) p. 9, line 12: The numerical estimation is described here as a bootstrap, but the method used is not the bootstrap (Efron and Tibshirani, 1986), but rather an ad-hoc procedure. I am puzzled here because the bootstrap calculation is very simple (replicate data sets produced by random sampling with replacement of the original data set), and it has well defined properties for estimation of uncertainties. In contrast, I have no idea if the ad-hoc procedure used here (randomly removing 25% of the data) is able to provide reliable estimates of uncertainties.

**We agree that a bootstrap method should not be removing but replacing samples; we have fixed this.**

5) p. 9, line 27: It would help to explain here why the closure temperature for Ar muscovite is cited here, given that this information is not used in the paper.

**We have removed the reference to the closure temperature, which, we agree, is irrelevant here.**

6) p. 12, fig. 4: The horizontal axes have no tic values or axis labels, and the vertical axes are also unlabeled.

**We have improved this figure**

7) p. 13, lines 5-9: The estimation method seems to be rather unstable.

**Using synthetic ages distributions we have shown the reliability of the method.**

8) p. 14, figure 5: This figure is hard to understand. It is my guess that the gray scale represents, not the estimated erosion rate, but rather the estimated relative erosion rate. Is that correct?

**The figure has been modified to be clearer.**

**Associate Editor's comments**

I find the manuscript basically acceptable as is. It is of fundamental interest and potentially of broad applicability. I am far from understanding the math details underlying the approach presented, but I trust the authors and future users of the method to deal with this if needed.

I have a few comments/questions that the authors are free to consider.

"The first piece of information comes from the ages themselves: catchments or sub-catchments where the proportion of grains with young ages dominates are likely to experience rapid exhumation today or in the recent past; whereas catchments or sub- catchments where the proportion of grains with old ages dominates are more likely to have experienced rapid erosion in a more distant past."

=> Why would "a catchment with old ages" be interpreted as representing an area of rapid erosion in the past. I thought old ages meant slow erosion. Are you only saying that pulses of rapid erosion can't be resolved by thermochronometric method when ages are old? Hence only catchment with young ages would be able to decipher rapid erosion. Isn't there some bias here?

**We have changed the sentence to reflect the point made by the AE.**

"For this, ages can be regarded as passive markers (or colors) that inform us on the proportion in which the mixing takes place today, which is directly proportional to

the present-day erosion rate."

=> what is the relation that determines a direct proportionality link between the "mixing of passive markers in a river" and "present-day erosion rate". Is this obvious or are there any references to back this up?

**Because faster present-day erosion rate will yield a proportionally larger sediment flux into the river, everything else being considered the same. We have adapted the sentence to avoid the confusion caused to the AE.**

"we have devised a simple method that, unlike many others such as that of Brewer et al. (2006), is only dependent on the raw, binned age data." I guess I understand that here you're bypassing the need to model individual age data into cooling rates through assumptions of geothermal gradients etc. . . If that's correct I must say that for a non-specialist it would be great to have a bit more material here on the assumptions that are made and not made.

**We have changed the text to make this clearer**

Also, are the ages really "raw" or do they come with uncertainty/standard error on them? And if yes, what are the uncertainties/SE on "raw" ages.

In the abstract it is said "We show that detrital age distributions contain dual information about present-day erosion rate" but in the text it is more an assumption than a demonstration. And I also failed to see how the results obtained are confronted with existing constraints on erosion rates in this area.

**We have added a paragraph comparing the distribution of erosion rates predicted by our method to previous estimates.**

4. Uncertainty estimates. Since the distribution are not normal, does it makes sense to use the standard deviation around the mode? Also, wouldn't it be possible to perform a standard error propagation that would include the standard errors on ages?

**We now show how well the method behaves when applied to synthetic ages distributions with finite age uncertainty. This clearly helps identifying the main sources of uncertainty.**

Conclusion: thanks to the authors for submitting their work to eSurf and apologies again for the slow process.

[revised manuscript text omitted]
\alpha_i \Big/ \sum{}^m \prod_{j=1}^{i-1} A(1+\delta_j\alpha_j{}^m) \quad \text{for } i=1, \frac{H_{i-1}^k - H_i^k}{H_i^k}\cdots, \frac{H_i^k - H_{i-1}^k}{1-H_i^k} M \tag{17}$$

 assuming that $\delta_1^m = 0$ and $\epsilon_1^m = 1$ such that the estimated minimum erosion rates are relative erosion rates scaled by the unknown erosion rate in the first catchment,

$$\epsilon_i = \frac{\delta_i}{A_i\alpha_i}\sum_{j=1}^{i-1} A_j\alpha_j\epsilon_j$$

From the values of the minimum contribution factors, $\delta_i^m$, we can also estimate the relative surface concentrations of each age bin $k$ in  each catchment $i$, using:

$$C_i^k = \frac{H_i^k - H_{i-1}^k}{\delta_i}\frac{H_i^k - H_{i-1}^k}{\delta_i^m} + H_i^k \tag{18}$$

**3 Using age distributions from tributaries**

Age distributions from tributaries can be included to improve the solution locally, i.e. in the  tributary catchment. Let's call $A_T$, $\alpha_t$ and $\epsilon_T$ the catchment area, the  mineral concentration factor and the mean erosion rate of the  tributary catchment, and $A_M$, $\alpha_M$ and $\epsilon_M$ the

catchment area, the  mineral concentration factor and the mean erosion rate of the rest of catchment $A_{\overline{i}}i$.

For each bin $k$ in the catchment $i$, we can write:

$$\underline{A}F_i\epsilon_i\underset{\sim}{\alpha_i}{}^m C_i^k = \underline{A}F_T\epsilon_T\underline{\alpha_T}C_T^k + \underline{A}F_M\epsilon_M\underline{\alpha_M}C_M^k \tag{19}$$

5  where $\underset{\sim}{F_T = A_T\alpha_T}$ and $\underset{\sim}{F_M = A_M\alpha_M}$. By conservation of eroded rock mass, we have:

$$\underline{A}F_M\epsilon_M\underline{\alpha_M} = \underline{A}F_i\epsilon_i\underline{\alpha_i} - \underset{\sim}{A^m} - F_T\epsilon_T\underline{\alpha_T} \tag{20}$$

which we can use to transform Equation (19) into:

$$\underline{A}F_i\epsilon_i\underset{\sim}{\alpha_i}{}^m C_i^k = \underline{A}F_T\epsilon_T\underline{\alpha_T}C_T^k + (\underline{A}F_i\epsilon_i\underline{\alpha_i} - \underset{\sim}{A^m} - F_T\epsilon_T\underline{\alpha_T})C_M^k \tag{21}$$

to obtain:

10  $$C_M^k = \frac{A_i\epsilon_i\alpha_i C_i^k - A_T\epsilon_T\alpha_T C_T^k}{\underline{A_i\epsilon_i\alpha_i - A_T\epsilon_T\alpha_T}}\frac{F_i\epsilon_i^m C_i^k - F_T\epsilon_T C_T^k}{\underset{\sim}{F_i\epsilon_i^m - F_T\epsilon_T}} \tag{22}$$

Using the method for the main trunk data described in the previous sections, we know $\epsilon_i$ and $C_i^k$. The tributary data (age distributions) gives us the $C_T^k$ ( ) and we can solve for the surface concentrations $C_M^k$ assuming first that the erosion rate is uniform in the catchment $i$, i.e. $\epsilon_T = \epsilon_M = \epsilon_i$, to give:

15  $$C_M^k = \frac{A_i\alpha_i C_i^k - A_T\alpha_T C_T^k}{\underline{A_i\alpha_i - A_T\alpha_T}}\frac{F_i C_i^k - F_T C_T^k}{\underset{\sim}{F_i - F_T}} \tag{23}$$

However, this may lead to unrealistic values of the relative surface concentrations $C_M^k$, i.e. not comprised between 0 and 1. Consequently, two conditions need to be added so that $0 < C_M^k < 1$ for all $k$ . The first condition ($C_M^k > 0$) yields:

$$\epsilon_T < \frac{A_i\alpha_i C_i^k}{\underline{A_T\alpha_T C_T^k}}\frac{F_i C_i^k}{\underset{\sim}{F_T C_T^k}}\epsilon_i{}^m \tag{24}$$

20  while the second condition ($C_M^k < 1$) yields:

$$\epsilon_T < \frac{A_i\alpha_i(1-C_i^k)}{\underline{A_T\alpha_T(1-C_T^k)}}\frac{F_i(1-C_i^k)}{\underset{\sim}{
[revised manuscript text omitted]